# Representing improved tropospheric ozone distribution over the Northern Hemisphere by including lightning NOx emissions in CHIMERE

Sanhita Ghosh[1], Arineh Cholakian[1], Sylvain Mailler[1, 2], and Laurent Menut[1]

[1]Laboratoire de Météorologie Dynamique, IPSL, École Polytechnique, Route de Saclay, Palaiseau, France, 91128
[2]École des Ponts, Marne-la-Vallée, Institut Polytechnique de Paris

**Correspondence:** Sanhita Ghosh (ghosh.sanhita@lmd.ipsl.fr; sanhitaghosh027@gmail.com)

**Abstract.** Estimating nitrogen oxide emissions from lightning (LNOx) in models is highly uncertain, affecting the accuracy of atmospheric composition and air quality assessments. Still, it is essential to include these emissions in models to increase the realism in representing the gases and aerosols. LNOx emissions have recently been incorporated into the updated version of the CHIMERE model (v2023r2). In this study, we evaluate the present state of modelling the lightning flashes over the Northern Hemisphere (NH), using a classical scheme based on cloud top height (CTH) and an updated ice flux-based scheme (ICEFLUX). We conduct a comprehensive 3D comparison of model outputs, including in situ measurements and satellite data, to rigorously assess the robustness and applicability of these parameterizations. The comparative analysis reveals that the CTH scheme provides a more accurate spatial variability of lightning flashes over lands and tropical oceans. Both parameterizations accurately capture the magnitude of lightning flashes over the tropics, while the ICEFLUX scheme is more effective in representing mid-latitudinal flashes. However, both schemes perform well in capturing the seasonal variation of lightning flashes. The estimated flash frequencies over the NH from the experiments closely align with satellite observations, and the LNOx emissions fall within the range reported in previous modelling studies. There is an overall increase in ozone ($O_3$) concentration due to inclusion of LNOx, which substantially improves the tropospheric $O_3$ distribution, specifically at the tropical free troposphere. The LNOx emissions hence critically influence the $O_3$ burden as well as the hydroxyl radicals, which further impact the atmospheric lifetime of trace gas methane.

## 1 Introduction

Nitrogen oxides (NOx), consisting of nitric oxide (NO) and nitrogen dioxide ($NO_2$), are trace gases that play a key role in atmospheric chemistry, particularly in the formation of tropospheric ozone ($O_3$) (Finney et al., 2014; Luo et al., 2017; Akimoto and Tanimoto, 2022). NOx emissions arise from both anthropogenic sources, e.g., fossil fuel combustion, biomass burning, and natural processes, such as lightning and soil-NOx emissions (Verma et al., 2021; Butler et al., 2020). Among these sources, lightning-induced NOx (LNOx) contributes approximately 10%–15% to global NOx emissions, with an even greater contribution (up to 70%) in the upper troposphere (Maseko et al., 2021; Luhar et al., 2021; Wu et al., 2023). Importantly, LNOx

has a stronger impact on tropospheric $O_3$ formation compared to surface-based sources, due to the altitude at which LNOx is injected into the atmosphere and the efficiency of $O_3$ production in the upper troposphere (Finney et al., 2016a; Luhar et al., 2021). The mean estimated rate of NOx emissions from lightning is highly uncertain, with recent studies indicating variations ranging between 33–660 moles NO per flash (Luhar et al., 2021; Bucsela et al., 2019; Murray, 2016; Schumann and Huntrieser, 2007); although Schumann and Huntrieser (2007) suggest a mean value of 250 moles NO per flash. In spite of this uncertainty, inclusion of these emissions in models is essential to enhance the accuracy and reliability of model projections.

The inclusion of LNOx in chemistry-transport models has been a research topic for several decades (Kang et al., 2020), with seminal studies by Price and Rind (1992); Price et al. (1997a); Schumann and Huntrieser (2007); Allen et al. (2010); Finney et al. (2014), pioneering the quantification of lightning flash rates and their associated NOx production. These foundational studies laid the groundwork for understanding the contribution of LNOx to tropospheric chemistry (Allen et al., 2010; Banerjee et al., 2014; Finney et al., 2016a; Kang et al., 2019, 2020). A range of parameterization schemes, including diverse empirical equations, have been developed over decades to quantify lightning flash rates and their spatial distribution (Finney et al., 2014). Despite the significant progress made, challenges remain in accurately quantifying LNOx emissions, due to uncertainties in characterizing both the spatial and temporal variations in lightning frequency and intensity, the apportionment among 'cloud to ground (CG)' and 'in cloud (IC)' flashes, the rates of NOx production from lightning discharges, as well as the vertical distribution and transportation of LNOx after its generation (Labrador et al., 2005; Schumann and Huntrieser, 2007; Menut et al., 2020a; Wu et al., 2023). Recent studies have focused on improving the representation of lightning in models through various parameterization schemes based on cloud top height (CTH, Price and Rind, 1992; Price et al., 1997a; Clark et al., 2017), ice flux (ICEFLUX, Finney et al., 2014), convective precipitation, updraft of mass flux (Allen et al., 2000; Allen and Pickering, 2002) and convective available potential energy (CAPE, Choi et al., 2005; Zhao et al., 2009). These approaches aim to better capture the spatial and temporal variability of lightning activity, leading to more accurate estimates of LNOx emissions.

In this study, we expand on the previous work by implementing the ICEFLUX scheme in chemistry-transport model CHIMERE and comparing it with the CTH scheme. The CHIMERE model has been developed since 1997 and modified on a regular basis for better prediction of trace gases and aerosols (Menut et al., 2020b). The improvement in the natural emissions in the recent version of the model, allows the incorporation of LNOx emissions (Menut et al., 2024). The study by Menut et al. (2020a), conducted over a short period of two months (July–August, 2013), covering Europe and the northern part of Africa, demonstrates changes in tropospheric $O_3$ and NOx concentrations resulting from the inclusion of LNOx emissions in CHIMERE, using CTH scheme. However, opportunities remain to improve the representation of flash rates in the model. To address this, we have applied the recent ICEFLUX parameterization. The CTH scheme does not incorporate the complex interactions and charge distributions that drive lightning production (Price and Rind, 1992; Finney et al., 2014), nor does it account for detailed microphysical processes, storm dynamics, and the presence of ice particles (Price and Rind, 1992). The improved modelling of cloud ice has facilitated the inclusion of the upward flux of ice crystals (Finney et al., 2014). However, ice flux alone is insufficient to fully capture the complexities of lightning phenomena, as additional factors likely influence the charging process. Therefore, a comparative analysis of the traditional CTH and the updated ICEFLUX schemes is essential to

assess their effectiveness in the regional model CHIMERE. We perform a 3D comparison of model outputs with each other and with a simulation devoid of LNOx. Model outputs are also compared with in situ measurements and satellite data. Furthermore, validating and evaluating these lightning parameterizations across different models (global and mesoscale) are crucial for fully assessing their robustness and applicability, underscoring the significance of this study. Additionally, a thorough evaluation of simulated tropospheric $O_3$ is also essential to refine model accuracy and deepen our understanding in the role of LNOx in atmospheric composition.

Lightning-generated NOx also influences the tropospheric hydroxyl radical (OH) budget, in addition to affecting $O_3$ concentrations (Murray et al., 2013; Murray, 2016; Mao et al., 2021). The OH radical is primarily formed due to photolysis of $O_3$ (O($^1$D)) at a shorter wavelength ($\leq$330 nm) in the presence of water vapour and secondarily through the reaction between hydroperoxyl radical ($HO_2$) and NO (Lelieveld et al., 2016; Banerjee et al., 2014). As a highly reactive and short-lived oxidant, with a lifetime of just a few seconds, OH is essential to tropospheric chemistry (Lelieveld et al., 2016). However, substantial variability exists among global models, with differences of up to $\pm$30% in estimating the mean OH burden (Murray et al., 2021). OH further controls the lifetime of many important trace gases, such as methane ($CH_4$), carbon monoxide (CO) and non-methane volatile organic compounds (NMVOCs) (Akimoto and Tanimoto, 2022; Luhar et al., 2021). For example, increase in OH burden reduces the lifetime of $CH_4$ (Equation R1), a potent greenhouse gas and a major contributor to global warming (Naik et al., 2013; Banerjee et al., 2014; Murray et al., 2021). By improving the parameterization of LNOx in CHIMERE, this study strengthens our understanding of tropospheric chemistry and the dynamics of trace gases.

$$CH_4 + OH \rightarrow CH_3 + H_2O \tag{R1}$$

Hence the specific objectives of the study are, (i) to evaluate and improve the modelling of lightning flashes, with the CHIMERE model using the classical CTH and the upgraded ICEFLUX scheme; (ii) to evaluate the effect of LNOx emissions on tropospheric $O_3$ and trace gases; (iii) the influence on the OH burden and lifetime of $CH_4$ quantified against the chemical loss. The detailed methodology is provided in Section 2. An analytical evaluation of the simulated results have been carried out and presented in the following sections.

## 2 Method of study

### 2.1 CHIMERE model configuration and experimental set-up

In this study, simulations are carried out with the CHIMERE chemistry-transport model (version 2023r2; Menut et al., 2024) over the domain of Northern Hemisphere (NH) expanded from 0°–90°N, at a horizontal resolution of 100×100 km$^2$. Here, meteorological fields are forced externally to CHIMERE with a 3-hourly forecast dataset from European Centre for Medium-Range Weather Forecasts (ECMWF)– Integrated Forecasting System (IFS) (https://www.ecmwf.int/en/forecasts/datasets, last access: 16 May, 2024). Simulations are done in twenty vertical levels in $\sigma$-pressure coordinates ranging from surface to 200

hPa for a period of one year (January–December, 2018) with a spin-up time of 15 days. The MELCHIOR2 scheme is used for the chemical mechanism. The CHIMERE model employs a 10-bin logarithmic sectional size distribution ranging from 0.01 to 40 $\mu$m. Fields of chemical concentration are calculated with a time-step of few minutes, using an adaptive time-step, to ensure that the Courant-Friedrichs-Lewy (CFL) stability criterion is satisfied (Menut et al., 2021). Boundary and initial conditions are derived from Copernicus Atmosphere Monitoring Service (CAMS) reanalysis dataset of atmospheric compositions produced by ECMWF, consisting of three-dimensional time-consistent atmospheric composition fields, including aerosols and chemical species (Inness et al., 2019), and from GOCART for dust concentrations (Chin et al., 2002). Biogenic emissions are provided by a reduced online version of the Model of Emissions of Gases and Aerosols from Nature (MEGAN) model (version 2.10) (Guenther et al., 2012). Mineral dust and sea-salt emissions are calculated using the schemes of Alfaro and Gomes (2001) and Monahan (1986), respectively. Anthropogenic emissions and fire emissions in the model are incorporated respectively from CAMS-global and CAMS Global Fire Assimilation System (GFAS, https://atmosphere.copernicus.eu/global-fire-emissions, last access: 16 May, 2024). The formation of secondary organic aerosols (SOAs) is as described in Pun and Seigneur (2007) and Bessagnet et al. (2008). The aerosol dynamic processes, such as condensation, coagulation, wet and dry deposition, absorption, and scavenging, are incorporated into the model (Menut et al., 2021). The mixing state is considered as internal homogeneous aerosol mixing (Menut et al., 2013). The online calculations for radiation and photolysis are incorporated using the FastJX module (Wild et al., 2000; Mailler et al., 2016). The horizontal and vertical transports are solved with the van Leer (1977) scheme. Boundary layer height and vertical diffusion are calculated by the parameterization proposed by Troen and Mahrt (1986) and deep convective fluxes are estimated using the Tiedtke (1989) scheme. Gaseous and aerosol species undergo dry or wet deposition and fluxes are calculated using the Wesely (1989) and Zhang et al. (2001) parameterization schemes. With access to anthropogenic and biogenic emissions, CHIMERE simulates 3D concentration for a range of gaseous and size-resolved particulate species, based on the chosen chemical scheme.

Simulations carried out for this study are (i) not including LNOx emissions (experiment: noLNOx), (ii) including LNOx emissions estimated with parameterization based on cloud top height (CTH) (experiment: LNOx-CTH) and (iii) LNOx emissions estimated with parameterization based on ice flux (experiment: LNOx-ICEFLUX).

## 2.2 Parameterization of lightning flash

### 2.2.1 Cloud top height-based parameterization (CTH)

Derived from the theories advanced by Vonnegut (1963) and Williams (1985), Price and Rind (1992) formulated the CTH parameterization, wherein the flash rate is contingent upon the cloud top height ($H_{top}$). The distinct relationships governing flash rates over land and ocean are delineated as follows:

$$F_l = a \times H_{top}^{4.9}$$
$$F_o = b \times H_{top}^{1.73}$$

(1)

Here, a and b are constants (values are provided in Table 1), $H_{top}$ represents the cloud top height above the ground level in km. Cloud-top height ($H_{top}$) and bottom height ($H_{bottom}$) are estimated based on the convection scheme of the model for each time step. F denotes the total flash rate in flash number $min^{-1}$ 25 $km^{-2}$, with subscripts 'l' and 'o' indicating land and ocean, respectively (Menut et al., 2020a). The distinction between land and ocean is employed to incorporate the disparity in updraft velocity over these two surface types. For instances where the cloud depth ($H_{top} - H_{bottom}$) is less than 5 km, the flash number is set to zero. This threshold reflects the physical condition necessary for charge separation and buildup in a storm to generate lightning. The assumption of minimum required cloud depth of 5 km, may introduce uncertainty in estimating lightning flashes, as it inherently assumes that every convective cloud with depth of 5 km corresponds to a thunderstorm (Luhar et al., 2021). It would be worthwhile to investigate the sensitivity of the modelled flash rates to the minimum cloud depth by varying this arbitrary threshold, either increasing or decreasing it. It's noteworthy that Price and Rind (1994) formulated an equation to adapt the above equations to various model resolutions. The scaling factor (C) determined to accommodate the model grid cell size is outlined as follows:

$$C = 0.97241e^{0.048203 \times \Delta x \times \Delta y} \tag{2}$$

Here, the product of longitude and latitude resolution, denoted as $\Delta x \times \Delta y$, is measured in degrees$^2$. This factor typically remains close to 0.97, and its impact on the results is minimal, especially when compared to the uncertainties due to other parameters. These uncertainties are offset by adjustment factors, that align the model more closely to observations (Gordillo-Vázquez et al., 2019). In preliminary simulations, we observed a highly overestimated flash rate, estimated based on the formulations by Price and Rind (1992), compared to the measurements from Lightning Imaging Sensor on the International Space Station (ISS-LIS) for the year 2018, over the land grids, followed by the ocean grids. Considering the overestimation in the modelled flash rate, we have applied factors 0.1 and 0.5, respectively to constants 'a' and 'b' in Equation 1 over the land and ocean grids, in experiment LNOx-CTH, to reconcile the modelled lightning flash rate to the satellite observations (Table 1).

**Table 1.** Values of constants in Equation 1

| constants | Price and Rind (1992) | present study (experiment: LNOx-CTH) |
|---|---|---|
| a | $3.44 \times 10^{-5}$ | $3.44 \times 10^{-6}$ |
| b | $6.40 \times 10^{-4}$ | $3.20 \times 10^{-4}$ |

For each grid-cell, the relative fraction of sea ($x_{sea}$) is determined using the land-sea mask from the land use database (Menut et al., 2020a). The total flash rate ($F_{CTH}$) is then calculated as follows:

$$F_{CTH} = \frac{C \times (x_{\text{sea}} \times F_o + (1 - x_{\text{sea}}) \times F_l)}{25} \tag{3}$$

### 2.2.2 Ice-flux-based parameterization (ICEFLUX)

145 The equations used to estimate the flash rates in ICEFLUX parameterization are as follows (Finney et al., 2014):

$$f_l = 6.58 \times 10^{-7} \phi_{ice}$$
$$f_o = 9.08 \times 10^{-8} \phi_{ice}$$

(4)

here, $f_l$ and $f_o$ represent the flash rate (flash number $m_{cell}^{-2}$ $s^{-1}$) over the lands and ocean, respectively. $\phi_{ice}$ denotes the upward ice flux ($kg_{ice}$ $m_{cloud}^{-2}$ $s^{-1}$) at 440 hPa and is determined using the following equation:

$$\phi_{ice} = \frac{q \times \Phi_{mass}}{c}$$

(5)

150 In this context, q represents the specific cloud ice water content at 440 hPa ($kg_{ice}$ $kg_{air}^{-1}$), $\Phi_{mass}$ denotes the updraft mass flux at 440 hPa ($kg_{air}$ $m_{cell}^{-2}$ $s^{-1}$), and c represents the fractional cloud cover at 440 hPa ($m_{cell}^{2}$ $m_{cloud}^{-2}$). Instances where c is less than 0.01 $m_{cloud}^{2}$ $m_{cell}^{-2}$, upward ice flux is set to zero. Additionally, if no convective cloud top is identified, the flash rate is also set to zero (Finney et al., 2016a). The total flash rate ($F_{ICEFLUX}$) is then calculated as follows:

$$F_{ICEFLUX} = x_{\text{sea}} \times f_o + (1 - x_{\text{sea}}) \times f_l$$

(6)

155 The estimated flash frequency from LNOx-ICEFLUX has been scaled down by a factor of 5 to align with satellite-observed frequencies. Consequently, the evaluation of LNOx-ICEFLUX results has been carried out using these adjusted flash rates.

### 2.2.3 Distribution of CG and IC lightning flashes

The empirically derived formula used to determine the relative proportion of 'cloud to ground (CG)' flashes in a single thunderstorm is initially based on the cold cloud depth ($H_f$, from 0°C to cloud top) (Price and Rind, 1993). In this study $H_f$ (in
160 km) is calculated with the temperature profile in the model, estimating the freezing temperature height or the freezing level. The modelled $H_f$ therefore varies from 6.9 to 7.76 km. The freezing level (0°C temperature), estimated in model, varies from 1 to 4.9 km, being the highest at tropics and decreasing with higher latitudes. The flashes from freezing level to the cloud top height is considered as 'in cloud (IC)' flashes and that from freezing level to ground as CG flashes. The freezing level acts as a natural boundary between the upper and lower parts of the cloud. Above the freezing level, ice particles contribute to the
165 development of IC lightning, while below it, the atmosphere is typically in a liquid state, with the warmer environment aiding in the development of CG lightning (Dwyer and Uman, 2014).

The mean ratio of IC to CG flash rates ($\beta$ = IC/CG) is estimated as 3.09 from model estimates. $\beta$, in this study, is estimated with the empirical equation (Equation 7) by Price and Rind (1993), which is frequently used in several modelling studies (Luhar et al., 2021; Gordillo-Vázquez et al., 2019). The empirical relationship between $\beta$ and the cold cloud depth ($H_f$) was developed by Price and Rind (1993) based on data collected for 139 individual thunderstorms over the western United States (US) during summer. Several studies support the fact that the parameterization Price and Rind (1993) successfully estimate the distribution of CG and IC flashes in global as well as in mesoscale models (Pickering et al., 1998; Fehr et al., 2004). Theoretically, $\beta$ varies between 1 to 50 for $H_f$ varying between 5.5 to 14 km to prevent unrealistic values. The value of $\beta$ estimated in our study is comparable to that obtained in recent modelling studies (Luhar et al., 2021; Gordillo-Vázquez et al., 2019). Wu et al. (2023) estimates the values of $\beta$ as 2.94–3.70 with a lightning nitrogen oxides (LNOx) emissions model using satellite-observed lightning optical energy. Further, experiments conducted with satellite- and ground-based observations over different parts of the world also produce a $\beta$ value in the range of 2.64–2.94 ($\pm$1.1–1.3) over US (Boccippio et al., 2001), 3–4 over India and China (Ghosh et al., 2023; Ren et al., 2024). $\beta$ = IC/CG (Equation 7), obtained in our study again shows consistency with the above mentioned results.

$$\beta = 0.021H_f^4 - 0.648H_f^3 + 7.49H_f^2 - 36.54H_f + 63.09 \tag{7}$$

The relative part of CG in the total (IC + CG) is denoted by $p$ (Equation 8). The estimated value of $p$ from our study is 0.25, aligning with findings from recent research (Luhar et al., 2021).

$$p = \frac{1}{1+\beta} \tag{8}$$

Hence, the $\beta$ in our study is not predetermined but is calculated based on the cold cloud thickness ($H_f$), estimated with temperature profile in CHIMERE and agrees well with the values estimated theoretically and from other model-based studies.

## 2.3 Estimation of LNOx emissions

Lightning flash energy estimates span a broad range from 0.35 to 5 GJ based on length-specific discharge values to up to 6.7 GJ considering contributions to the global atmospheric electric circuit (Krider et al., 1968; Uman, 2001; Price et al., 1997a). The NOx production rate per unit discharge energy also exhibits substantial variation, ranging from $1.1 \times 10^{16}$ to $50 \times 10^{16}$ molecules J$^{-1}$ in laboratory experiments (Schumann and Huntrieser, 2007), and $5$–$15 \times 10^{16}$ molecules J$^{-1}$ in theoretical models (Price et al., 1997a). In this study, we adopt flash energies of 3 GJ for CG flashes and 0.9 GJ for IC flashes, along with a NO production rate of $14.2 \times 10^{16}$ molecules NO J$^{-1}$ (Schumann and Huntrieser, 2007). Using these values, we calculate

the NO production in moles per flash as described in Equation 9, yielding a mean value of 332 moles of NO per flash. This estimation considers CG flashes as 25% of the total lightning flashes.

$$P(\text{CG, NO}) = 697.44 \, \text{moles flash}^{-1},$$
$$P(\text{IC, NO}) = 199.27 \, \text{moles flash}^{-1}$$

(9)

Recent research indicates comparable NO production rates for CG and IC lightning flashes, with a mean of 70–700 moles NO per flash (Bucsela et al., 2019; Ott et al., 2010; Finney et al., 2016a; Luhar et al., 2021). Despite this, significant differences in NOx production between IC and CG flashes are well-documented through theoretical models and observational studies, emphasizing the challenges and variability in quantifying NOx production rates (Gordillo-Vázquez et al., 2019; Carey et al., 2016; Koshak et al., 2014; Pickering et al., 1998; Price et al., 1997a). Global modelling efforts, such as those using National Aeronautics and Space Administration (NASA) Goddard Earth Observing System (GEOS)-5 and GEOS-Chem systems, combined with satellite and airborne observations, used lightning NO production rates of 260 moles per flash (Jourdain et al., 2010), 246 moles per flash (Liaskos et al., 2015), and a range of 346 (over tropics) to 665 (for mid-latitude region) moles per flash (Nault et al., 2017). Additionally, Miyazaki et al. (2014) derived a global average of 310 moles NO per flash by integrating lightning data from the Optical Transient Detector (OTD) and the Lightning Imaging Sensor (LIS) with atmospheric composition measurements in a global chemistry-transport model. Overall, the mean NO production varies across 2 to 3 orders of magnitude in moles NO per flash, as noted by Bucsela et al. (2019); Murray (2016); Schumann and Huntrieser (2007), although a mean value of 250 moles NO per flash has been suggested by Schumann and Huntrieser (2007). The estimate derived in this study is close to the values estimated by Miyazaki et al. (2014); Luhar et al. (2021), consistent with prior findings, underscoring the complexity of accurately assessing lightning-induced NO production. Lightning generate $NO_2$ with $NO_2$/NOx ratio varying from 0.1 to 0.5 (Schumann and Huntrieser, 2007). Therefore, it is important to include $NO_2$ emissions also. The $NO_2$ emissions from lightning are assumed to be 10% of the NO emissions estimated due to lightning in this study.

## 2.4 Estimation of CH$_4$ lifetime due to chemical loss

The loss in tropospheric methane (CH$_4$) is primarily (90%) due to oxidation by hydroxyl radicals (OH, Reaction R1)(Ghosh et al., 2015). The estimation of tropospheric chemical loss rate of CH$_4$ is as follows (in molecules cm$^{-3}$ s$^{-1}$) (Zhao et al., 2023):

$$\text{rate} = k(T)[\text{CH}_4][\text{OH}]$$

(10)

where, [CH$_4$] and [OH] are the concentrations of CH$_4$ and OH (in molecules cm$^{-3}$). [OH] is taken from the simulations in CHIMERE, whereas [CH$_4$] is from chemical boundary conditions derived from CAMS reanalysis dataset of atmospheric com-

positions, as $CH_4$ anthropogenic emissions are not taken into account in the model. The reaction rate (k(T) in $cm^3$ $molecule^{-1}$ $s^{-1}$) is temperature (T) dependent (Burkholder et al., 2019) and is represented in (Menut et al., 2013),

$$k(T) = 2.3 \times 10^{-12} \exp\left(-\frac{1765}{T}\right) \tag{11}$$

The total tropospheric chemical loss of $CH_4$ (L in Tg $yr^{-1}$) is estimated as,

$$L_{CH_4} = \int_V k(T)[CH_4][OH]\, dV \tag{12}$$

dV is the differential volume element in the troposphere. The lifetime of $CH_4$ ($\tau_{CH_4}$ in year) is expressed as,

$$\tau_{CH_4} = \frac{B_{CH_4}}{L_{CH_4}} \tag{13}$$

Here, $B_{CH_4}$ is the annual tropospheric burden (in Tg) of $CH_4$. Note that, all the calculations in our study are done for NH.

## 2.5  Observation data for evaluation

Flash data for the year 2018 from Lightning Imaging Sensor (LIS) on the International Space Station (ISS) platform, is used for evaluating flash rate estimated with the model (http://ghrc.nsstc.nasa.gov/; last access: 5 July, 2024). ISS-LIS optically detects lightning flashes that occur within its field-of-view during both day and night with storm-scale (4 km $\times$ 4 km) horizontal resolution (Blakeslee et al., 2020) and 2 ms of temporal resolution. After time corrections comparing with Geostationary Operational Environmental Satellite (GOES) 16 and 17 Geostationary Lightning Mappers (GLM-16/17) and ground-based observations, the timing accuracy of ISS-LIS is better than its native precision of 2 ms. ISS operates in low Earth orbit (LEO) and overpasses one region on the earth surface up to three times a day and up to two times in the tropics. Lightning observation of a specific point lasts up to 90 seconds per overpass (Erdmann et al., 2020). The flash detection efficiency of ISS-LIS is around 60% with diurnal variability of 51%–75% (Blakeslee et al., 2020). Monthly averaged flash rates are obtained from combined climatology product of satellite observations from the Optical Transient Detector (OTD) and the Lightning Imaging Sensor (LIS), launched with the MicroLab-1 satellite, for the period of May, 1995 to December, 2014 (http://ghrc.nsstc.nasa.gov/; last access: 21 November, 2024). The product utilized in this study is the High Resolution Monthly Climatology (HRMC), which offers 12 monthly values at a horizontal resolution of 0.5° $\times$ 0.5°. Details are provided in Cecil et al. (2014).

For evaluating the vertical profile of $O_3$, altitudinal data measured by ozone-sonde, launched on small balloons, are downloaded from the World Ozone and Ultraviolet Radiation Data Centre (WOUDC, https://woudc.org/data, last access: 5 July, 2024) for the year 2018. Ozone-sonde data from 122, 977 and 121 stations are collected, respectively, over the tropical

(0°–30°N), mid-latitudes (30° N–60° N) and polar region (60° N–90° N). We also have used vertical $O_3$ profile data from Southern Hemisphere ADditional OZonesondes (SHADOZ) ozone-sonde measurements (https://tropo.gsfc.nasa.gov/shadoz, last access: 21 November, 2024) at four tropical stations (Kuala Lampur: 3.14°N, 101.69°E; Hanoi: 21.02°N, 105.80°E; Costa Rica: 9.62°N, −84.25°E and Hilo: 19.72°N, −155.08°E) for the year 2018. Global distributions of tropospheric column of ozone (TCO) are derived from the Ozone Monitoring Instrument (OMI) and Microwave Limb Sounder (MLS) on board the

Aura satellite (https://acd-ext.gsfc.nasa.gov/; last access: 5 July, 2024) for the year 2018. The monthly mean TCO data from OMI/MLS are derived by subtracting the stratospheric column of ozone (SCO) from the total column of ozone measured by the OMI sensor. This process utilizes the tropospheric ozone residual (TOR) algorithm along with stratospheric ozone profile information from the MLS sensor (Ziemke et al., 2006). The dataset covers the spatial range of ±60° with a spatial resolution of 1° × 1.25°, spanning the period from October 2004 to December 2020.

This study utilizes total tropospheric column $NO_2$ from daily global gridded (0.25° × 0.25°) $NO_2$ Product from OMI L3-level (https://data.gesdisc.earthdata.nasa.gov/; last access: 2 December, 2024) for the year 2018 (Levelt et al., 2018). OMI is an ultraviolet-visible (UV-Vis) spectrometer on the polar-orbiting NASA Aura satellite, launched on 15 July, 2004 (Lamsal et al., 2021). The simulated $O_3$ and $NO_2$ mixing ratio is compared with ground-based observations from OpenAQ (https://openaq.org; last access: 5 July, 2024; Hasenkopf et al., 2015), U.S. Environmental Protection Agency (EPA, https://www.epa.gov; last ac-

cess: 5 July, 2024), European Environment Agency (EEA, https://www.eea.europa.eu; last access: 5 July, 2024), Environment and Climate Change Canada (ECCC) data catalogue (https://data-donnees.az.ec.gc.ca, last access: 5 July, 2024), Subsistema de Información de Calidad del Aire (SISAIRE, http://sisaire.ideam.gov.co; last access: 5 July, 2024) and China National Environmental Monitoring Centre (CNEMC, https://quotsoft.net/air/; last access: 5 July, 2024; Dufour et al., 2021), collected over the study period. The total number of observation stations over the NH are mentioned in Table 6. The evaluation of simulated

data is done with the statistical analyses estimating the mean absolute bias (MAB), normalized mean error (NME), and root mean square error (RMSE), using the annual mean of $O_3$ and $NO_2$ mixing ratio.

## 3 Results and Discussions

### 3.1 Evaluation of modelled lightning flash rate

In this section we analyse the estimated lightning flash rates over the Northern Hemisphere (NH), from simulations conducted

with CHIMERE model applying parameterization schemes based on cloud top height (experiment: LNOx-CTH) and ice flux (experiment: LNOx-ICEFLUX). The modelled flash rates from the two simulations are compared with observed flash rates from ISS-LIS (domain: ±55° latitudes) for the year 2018 and the combined climatology product of satellite observations from the LIS/OTD for the period of May, 1995 to December, 2014. The spatial distribution of annual flash rates are presented in Figure 1. Please note that the flash rates over the land, estimated with ICEFLUX parameterization, are divided by 5 for each

grid to match the ISS-LIS satellite observations.

As observed from our study, warm tropical regions (0°–30°N), especially central Africa, south America, India and south China are the regions with high lightning flash rate due to large convective activity, followed by the mid-latitudes (30° N–

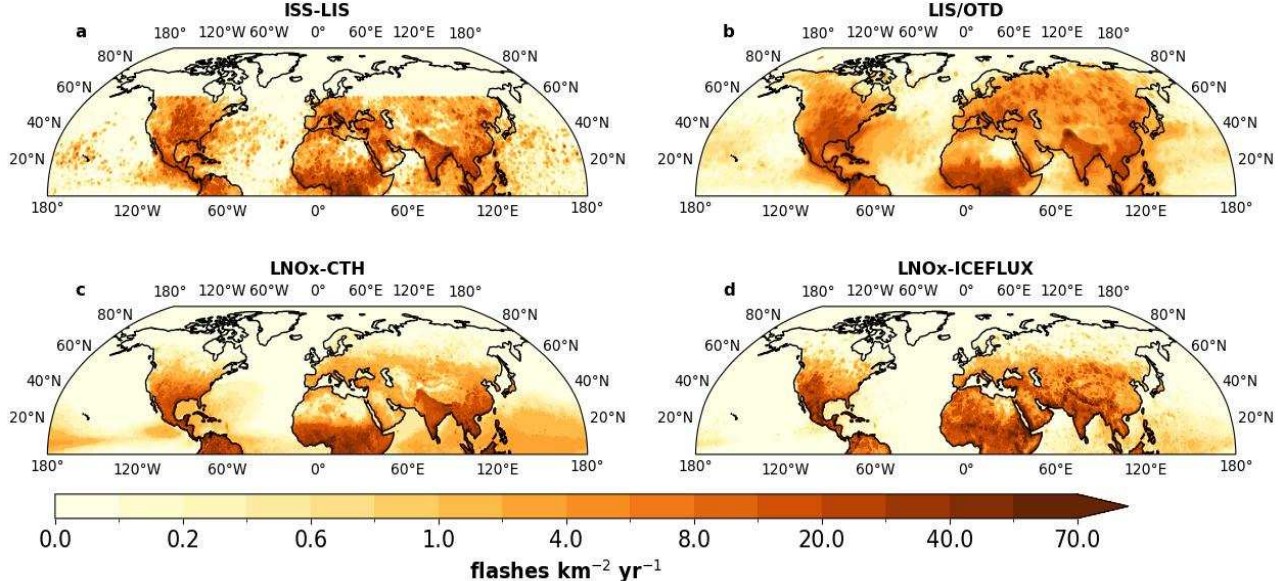

**Figure 1.** Spatial distribution of annual flash rates (flashes km$^{-2}$ yr$^{-1}$) over NH based on (a–b) observations from (a) ISS-LIS satellite for the year 2018 (domain: 0–55°N) and (b) LIS/OTD climatology data (May, 1995 to December, 2014), (c–d) simulation experiments (c) LNOx-CTH and (d) LNOx-ICEFLUX.

60°N; Figures 1c and 1d). South Asia, including India and south China, where significant flash rates are observed, are known to exhibit the greatest seasonal and interannual variation in lightning activities (Pawar et al., 2012a; Xu et al., 2023). The annual
flash rate is observed as 10–20 flashes km$^{-2}$ yr$^{-1}$ from both the experiments, over most of the tropical lands (Figures 1c and 1d), which is in agreement to the satellite observations from ISS-LIS and LIS/OTD (Figures 1a and 1b). Over the mid-latitudes, flash rates are observed in the range 2–4 flashes km$^{-2}$ yr$^{-1}$, while the polar regions (60° N–90°N) exhibit significantly lower values (0.1–0.2 flashes km$^{-2}$ yr$^{-1}$) as indicated by the simulations as well as the satellite observations. Therefore, the tropical and mid-latitude land regions dominate lightning activity. The spatial distribution of flash rates closely resembles the patterns
observed in previous model-based studies using CTH and ICEFLUX schemes (Luhar et al., 2021; Gordillo-Vázquez et al., 2019; Finney et al., 2014, 2016a; Murray, 2016). However, patches of high flash rate observed in satellite data over central Canada, central and south-eastern part of the United States, central European countries, and northern Russia are not reflected in the modelled flash rates from either experiment. Additionally, the elevated flash rates over central Asia are not captured in the LNOx-CTH simulation. Flash rates over land are significantly higher than the oceans due to intense convection over land
regions (Albrecht et al., 2016). In oceanic regions, relatively higher flash rates (1–2 flashes km$^{-2}$ yr$^{-1}$) are observed in the tropical regions, particularly over the Bay of Bengal and the Pacific Ocean, as simulated in LNOx-CTH experiment. Previous studies have reported inconsistencies in the equation for oceanic flashes developed by Price and Rind (1992) (Michalon et al., 1999; Boccippio, 2002; Luhar et al., 2021). However, our study demonstrates an improved flash rate distribution over tropical oceans, using the CTH scheme with a correction factor of 0.5 applied to constant 'b' in Equation 1, for the oceanic grids. A

simulation with the original scheme by Price and Rind (1992) showed an overestimated flash rates over the tropical ocean by a factor $\approx 2$, which is lowered to 1.3–1.4, after application of the correction factor in the present study. On the other hand, the magnitude of oceanic flash rates are significantly lower in simulations using the ICEFLUX scheme compared to the CTH scheme. The ICEFLUX scheme explicitly relates lightning flash rates to the upward ice flux; therefore, the weaker updraft strength in oceanic storms leads to less efficient charge separation, resulting in fewer lightning flashes over the ocean (Finney et al., 2014).

**Table 2.** Mean annual flash rate over NH, lands and ocean in NH for three latitude bands. Latitudinal coverage for the lightning data from ISS-LIS is $0°$–$55°$N.

| Name of the experiments/ satellite data | mean flash rates (flashes km$^{-2}$ yr$^{-1}$) | | | | | | | | |
|---|---|---|---|---|---|---|---|---|---|
| | 0°–30°N | 30°–60°N | 60°–90°N | 0°–30°N | 30°–60°N | 60°–90°N | 0°–30°N | 30°–60°N | 60°–90°N |
| | NH | | | Land in NH | | | Ocean in NH | | |
| LNOx-CTH | 4.69 | 0.86 | 0.18 | 10.95 | 1.46 | 0.55 | 0.75 | 0.2 | 0.0003 |
| LNOx-ICEFLUX | 4.55 | 1.54 | 0.36 | 12.45 | 3.92 | 1.1 | 0.37 | 0.1 | 0.0002 |
| ISS-LIS | 4.14 | 1.9 | | 11.13 | 2.7 | | 0.51 | 0.2 | |
| LIS/OTD | 4.2 | 2.7 | 0.37 | 12.2 | 4.1 | 1.24 | 0.55 | 0.27 | 0.05 |

**Table 3.** Statistical analysis of spatially varying annual flash rates from model comparing with ISS-LIS and LIS/OTD satellite observations over NH, land and ocean in NH for three latitude bands. Latitudinal coverage for the ISS-LIS data is $0°$–$55°$N. Statistical scores for comparison of simulated flash rates with LIS/OTD observations are provided in the parentheses.

| Name of the experiments | $r^{\perp}$ | | | RMSE (flashes km$^{-2}$ yr$^{-1}$)$^{\perp}$ | | | NME (%)$^{\perp}$ | | |
|---|---|---|---|---|---|---|---|---|---|
| | 0°–30°N | 30°–60°N | 60°–90°N | 0°–30°N | 30°–60°N | 60°–90°N | 0°–30°N | 30°–60°N | 60°–90°N |
| | NH | | | | | | | | |
| LNOx-CTH | 0.59 (0.73) | 0.53 (0.66) | (0.5) | 8.21 (6.64) | 4.20 (4.06) | (1.15) | 93.54 (65.28) | 80.37 (76.15) | (96.77) |
| LNOx-ICEFLUX | 0.31 (0.41) | 0.28 (0.36) | (0.26) | 12.23 (11.35) | 5.70 (5.50) | (1.15) | 119 (97.65) | 108.11 (90.67) | (96.92) |
| | Land in NH | | | | | | | | |
| LNOx-CTH | 0.68 (0.82) | 0.64 (0.63) | (0.3) | 11.65 (9.52) | 4.28 (5.15) | (3.22) | 61.41 (37.68) | 75.48 (74.34) | (96.88) |
| LNOx-ICEFLUX | 0.15 (0.21) | 0.27 (0.34) | (0.12) | 21.50 (21.33) | 9.34 (8.77) | (3.23) | 110.65 (92.57) | 154.87 (93.60) | (97.10) |
| | Ocean in NH | | | | | | | | |
| LNOx-CTH | 0.06 (−0.28) | 0.15 (0.63) | (−0.1) | 0.94 (0.61) | 0.41 (0.35) | (0.10) | 132.10 (87.88) | 101.65 (94.26) | (100) |
| LNOx-ICEFLUX | 0.06 (0.17) | 0.14 (0.15) | (−0.1) | 0.94 (0.69) | 0.42 (0.37) | (0.10) | 99.88 (98.88) | 100 (99.88) | (100) |

$^{\perp}$Correlation coefficient (r), RMSE and NME are estimated comparing simulated flash rates and ISS-LIS and LIS/OTD satellite observations for the spatially varying annual mean flash rates.

The spatial mean of annual flash rate over the NH tropics and NH tropical lands are comparable from two experiments, whereas, the flash rate from LNOx-CTH is lower than that from LNOx-ICEFLUX, by a factor of two at mid-latitudes (Table 2). These flash rates over tropics are 5–8 times higher than the mid-latitudes for NH and lands, as estimated from experiment with CTH. These factors are comparatively lower from experiment with ICEFLUX and from satellite observations, explaining the effectiveness of ICEFLUX scheme over CTH, in capturing flashes over the mid-latitudes. While CTH scheme provides an useful approximation, since deeper convection generally correlates with higher lightning activity, it likely doesn't capture the

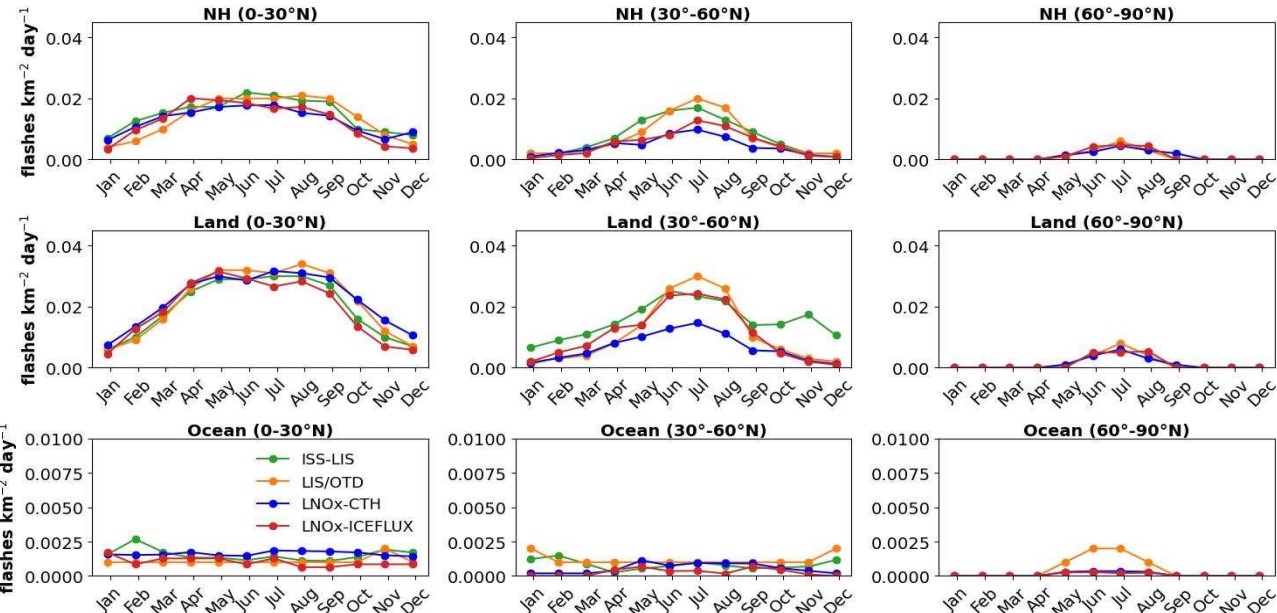

**Figure 2.** Comparison of monthly mean flash rates (flashes km$^{-2}$ day$^{-1}$) from simulations (LNOx-CTH and LNOx-ICEFLUX) with ISS-LIS satellite observations for the year 2018 (domain: 0–55°N) and LIS/OTD monthly climatology (May, 1995 to December, 2014), over NH, land and ocean in NH for the three latitude bands. Please note that the Y-axis in the plots for oceans is presented on a different scale.

full complexity of the processes driving lightning generation. Factors, such as updraft strength, cloud depth, ice water content and mixed-phase regions play critical roles in charge separation and lightning production. By strictly capping cloud heights at the tropopause in the CTH scheme, the model may indeed underestimate flash rates in the mid-latitudes. This highlights the

need to consider a multi-parameter approach for estimating flash rates, incorporating updraft dynamics, cloud microphysics and ice-phase processes alongside cloud top height. ICEFLUX scheme, by explicitly modelling ice fluxes, provides a more realistic approach in predicting charge separation and lightning activity (Finney et al., 2014). This leads to improved flash rate estimations, particularly in midlatitude storms where vertical motion, ice microphysics and latent heat fluxes play a complex role in thunderstorm electrification. Spatially mean annual flash rates over tropical lands from both the experiments are close

enough to the satellite measurements at tropics over NH and lands, while at mid-latitudes, only a good resemblance is observed for that from LNOx-ICEFLUX. On the other hand, the simulation with CTH scheme estimates the flash rates over tropical ocean as almost twice of that estimated using the ICEFLUX scheme, unlike that shown in previous studies (Finney et al., 2016b). We compare the spatially varying simulated annual flash rates with satellite observations (ISS-LIS and LIS/OTD) and the corresponding statistical metrics are presented in Table 3. Correlation coefficients for spatially varying flash rates show

comparatively stronger correlations over the tropics and mid-latitudes of the NH and land regions, between that from LNOx-CTH and satellite data, compared to LNOx-ICEFLUX, being consistent with the findings by Clark et al. (2017). The flash rates from LNOx-CTH exhibit significantly higher correlations, particularly when evaluated against LIS/OTD data. Analysis of RMSE and NME also indicates lower errors for LNOx-CTH over these regions in comparison to that observed for LNOx-

ICEFLUX, indicating the spatial variations of flashes from the first experiment align well with the satellite data. The worst performance is observed in the polar lands compared to the other two latitude bands, and also over oceanic region from both the experiments, characterized by weak correlations and higher errors. Hence, both schemes struggle to accurately simulate flash rates over oceans and high latitude lands. However, since lightning activity is minimal in these regions, the impact of this limitation is relatively minor. These results also underscore the ongoing challenges of accurately representing convection and capturing lightning flashes over the oceans. In summary, the statistical analysis points out the effectiveness of the LNOx-CTH scheme in reproducing the spatially varying lightning flashes reasonably well, particularly at tropics over NH and lands. However, both schemes exhibit limitations in the polar regions and over oceans, indicating scopes for further improvement in the parameterizations. While LNOx-ICEFLUX provides a reasonable estimate of flash rate magnitudes over both the tropics and mid-latitudes, it struggles to accurately capture the observed spatial pattern of lightning flashes, emphasizing the need for further improvement.

Figure 2 compares the monthly averaged lightning flash rates across different latitude bands in the NH, over land and ocean regions. The figure incorporates satellite observations (ISS-LIS and LIS/OTD) and results from two simulation experiments (LNOx-CTH and LNOx-ICEFLUX). Flash rates from the LNOx-CTH experiment show a clear seasonal cycle, with peaks occurring in May–August over NH and lands. During the winter months (November–January), flash rates drop significantly, being 5–7 times lower than the summer (May–August) peak values. The seasonal variation observed in the LNOx-ICEFLUX experiment and satellite observations closely align with this trend. Over land, both observations and simulations indicate high flash rates, particularly in the tropics, followed by mid-latitudes for all the months. Peak lightning activity over lands occurs during late spring and early summer (May–August), corresponding to enhanced convective activity (Holle et al., 2016; Ghosh et al., 2023). In contrast, flash rates over oceans are consistently lower over all latitude bands. The tropical ocean shows an uniform flash rates throughout the year, without any prominent seasonality. The delay in the seasonal peak of flash rates over NH with CTH, and particularly with ICEFLUX, as noted by Finney et al. (2014), is not seen in our simulations. Therefore, using near-real-time, high spatially and temporally resolved meteorological data from ECMWF-IFS with continuous updates and improved configuration for advection in CHIMERE, we achieve an improved seasonal distribution that match well the satellite measurements.

The modelled monthly mean flash rates from the simulations exhibit a strong positive correlation with satellite observations (ISS-LIS and LIS/OTD), with correlation coefficients ranging from 0.85 to 0.97 (Table 4). This agreement is consistent across all latitude bands over the NH, land regions, and the polar ocean region from both the experiments (LNOx-CTH and LNOx-ICEFLUX). These findings indicate that the simulations successfully capture the seasonal variability of flash rates. In contrast, a weaker negative temporal correlation is observed over tropical and mid-latitudinal oceans, indicating an inverse relationship between simulated and observed seasonality in flash rates in these regions. When comparing the simulated monthly flash rates with satellite observations, the results from LNOx-CTH align more closely with satellite observations at tropics than the mid-latitudes, as evidenced by lower NME at tropics over NH, land and ocean regions. Notably, the simulation LNOx-ICEFLUX exhibits a better performance at mid-latitudes than the LNOx-CTH, over NH and land regions, especially when compared with

**Table 4.** Statistical analysis of monthly mean flash rates from model comparing with ISS-LIS and LIS/OTD satellite observations over NH, land and ocean in NH for three latitude bands. Statistical scores for comparison of simulated flash rates with LIS/OTD observations are provided in the parentheses.

| Name of the experiments | r[‡] | | | NME (%)[‡] | | |
|---|---|---|---|---|---|---|
| | 0°–30°N | 30°–60°N | 60°–90°N | 0°–30°N | 30°–60°N | 60°–90°N |
| | NH | | | | | |
| LNOx-CTH | 0.86 (0.84) | 0.83 (0.94) | (0.97) | 17.4 (24.5) | 44.3 (43.1) | (34.6) |
| LNOx-ICEFLUX | 0.92 (0.85) | 0.89 (0.92) | (0.81) | 24.8 (22.4) | 33.6 (33.5) | (29.2) |
| | Land in NH | | | | | |
| LNOx-CTH | 0.92 (0.98) | 0.87 (0.97) | (0.97) | 16 (11.3) | 56.7 (41.1) | (23.5) |
| LNOx-ICEFLUX | 0.92 (0.93) | 0.85 (0.96) | (0.85) | 23.4 (16.9) | 31.5 (20.6) | (37.6) |
| | Ocean in NH | | | | | |
| LNOx-CTH | −0.1 (−0.28) | −0.48 (−0.49) | (0.87) | 27.5 (57.3) | 57.2 (53.7) | (78.3) |
| LNOx-ICEFLUX | 0.04 (−0.17) | −0.74 (−0.49) | (0.88) | 34.5 (32.9) | 76.2 (76.5) | (83.8) |

[‡]correlation coefficient (r) and NME estimated comparing simulated monthly mean flash rates and the same from ISS-LIS and LIS/OTD satellite observations.

**Table 5.** Estimated flash frequencies and LNOx emissions from simulations over NH. Flash frequencies in parentheses are estimated for the domain 0–55°N, comparable to the ISS-LIS satellite data (23.6 flash s$^{-1}$). The flash frequencies from LIS/OTD climatology data are 26.4 and 25.3 flash s$^{-1}$ over NH and for the domain 0–55°N.

| Name of experiments | flash frequency (flash s$^{-1}$) | total LNOx emissions (Tg N yr$^{-1}$) | correction factor |
|---|---|---|---|
| LNOx-CTH | 20.7 (20.64) | 2.8 | none |
| LNOx-ICEFLUX | 21.6 (21.53) | 3.1 | 5 |

LIS/OTD measurements. Further, advancement in the seasonal representation of oceanic convection processes is essential for improving the simulation of flash rates over oceans.

The annual flash frequencies over the NH is estimated as 20.7 and 21.6 flashes s$^{-1}$, respectively from the LNOx-CTH and LNOx-ICEFLUX experiments (Table 5). These values are consistent with the satellite observations (23.6 and 26.4 flashes s$^{-1}$, respectively from ISS-LIS and LIS/OTD observation over NH) as well as to that obtained in recent model-based studies over NH (Luhar et al., 2021). While no scaling factor is applied in the simulated flash rates from LNOx-CTH, the flash frequency from LNOx-ICEFLUX is divided by a factor 5 to reconcile with the satellite-observed frequency. Accordingly, the evaluation of the results from LNOx-ICEFLUX has been conducted using flash rates adjusted by this factor. For instance, a recent study by Finney et al. (2016a) utilizing the UK Chemistry and Aerosol (UKCA) model determined that the global flash rate scaling factors required for the UKCA model are 1.44 and 1.12 for the CTH and ICEFLUX lightning parameterizations, respectively. Another study by Gordillo-Vázquez et al. (2019) produce scaling factors 2.05 and 4, respectively for the CTH and ICEFLUX lightning schemes in Community Atmosphere Model (CAM5). A study by Tost et al. (2007) reported that scaling factors may vary by up to 2–3 orders of magnitude, depending on the lightning parameterization used and the resulting flash rate, to better match the observations. Uncertainties in the estimated lightning frequency may arise from the input meteorological data, model

configuration and the detection efficiency of satellite measurements is also a significant factor when comparing the modelled flashes with satellite observations (Blakeslee et al., 2020; Erdmann et al., 2020; Zhang et al., 2023).

## 3.2 NOx emissions from lightning

The production of LNOx emissions depends on the average flash frequency (in flashes $s^{-1}$) and the NOx production efficiency per flash, which represents the rate of NOx emissions per flash (Schumann and Huntrieser, 2007; Gordillo-Vázquez et al., 2019; Bucsela et al., 2019; Luhar et al., 2021). In this study, the mean LNOx emission per flash is estimated as 332 mol. The estimated annual NO emissions from lightning are 2.8 Tg N $yr^{-1}$ and 3.1 Tg N $yr^{-1}$ over the NH, respectively from LNOx-CTH and LNOx-ICEFLUX simulations (Table 5). Notably, LNOx emission from LNOx-ICEFLUX is 7.5% higher than

that from LNOx-CTH. Recent studies estimate global LNOx emissions typically range from 2 to 8 Tg N $yr^{-1}$ (Schumann and Huntrieser, 2007; Finney et al., 2016b; Nault et al., 2017), with variations reaching up to 25 Tg N $yr^{-1}$ in extreme scenarios (Price et al., 1997a, b). Price et al. (1997b) suggested that, the global annual LNOx emissions cannot be less than 5 Tg N or exceed 25 Tg N. The LNOx emissions from our study align well with the estimates by Luhar et al. (2021), which ranged from 2.39 to 3.41 Tg N $yr^{-1}$ over the NH when using CTH and a new parameterization by Luhar et al. (2021). A recent study

shows that the estimated global LNOx emissions, with CTH scheme and the new parameterization by Luhar et al. (2021), are respectively 5.66 and 5.58 Tg N $yr^{-1}$ in the model EMAC (Pérez-Invernón et al., 2024). These estimates are respectively, 17% higher and 15% lower than that estimated by Luhar et al. (2021) with the same parameterizations, showing the LNOx emissions are highly sensitive to the model configurations. The monthly variation in LNOx emissions, in our study, from the two simulations is shown in Figure 3a. The results indicate that peak emissions occur in July–August, followed by May–June

and September, in the LNOx-CTH experiment. The emissions from the LNOx-ICEFLUX experiment peak in May, followed by the remaining summer months (April, June–August). Approximately 60%–70% of the total annual LNOx emissions are contributed during late spring and summer (April–August), when lightning activity is at its highest.

The Figures 3b–3e represent the vertical distribution of LNOx emissions as percentage of LNOx mass per km. The emissions from CG and IC flashes are calculated separately considering CG flashes only below the freezing level and the IC flashes

only above the freezing level and below the cloud top. A simple vertical structure of the emissions is adopted in this study, considering the emissions to be evenly distributed over an altitude range. The distribution shows the maximum of LNOx mass lies between the altitude range of 4–7 km at all regimes from both the simulations, showing the typical 'backward C-shape' (Ott et al., 2010). 60–65% of LNOx mass is injected at this altitude range. Here it is to be mentioned that annually 1.85 and 1.9 Tg N LNOx are being generated over tropical land as obtained from LNOx-CTH and LNOx-ICEFLUX simulations, respectively,

being almost 63%–66% of total annual LNOx over NH. The amounts are 0.55 (0.76), 0.15 (0.09) and 0.07 (0.03) Tg N for mid-latitudinal land, tropical ocean and mid-latitudinal ocean respectively, from LNOx-CTH (LNOx-ICEFLUX) simulation. Therefore, the mid-tropospheric region (4–8 km) contributes the maximum of the LNOx mass, specially over the tropical land region. The vertical profiles available from previous studies, e.g., Pickering et al. (1998); Ott et al. (2010); Luhar et al. (2021), reveal a similar shape of all the profiles but contributing maximum at upper tropospheric region (within 2–4 km of the

tropopause) rather than mid-troposphere. However, the study by Pickering et al. (1998) represents a high emission near surface

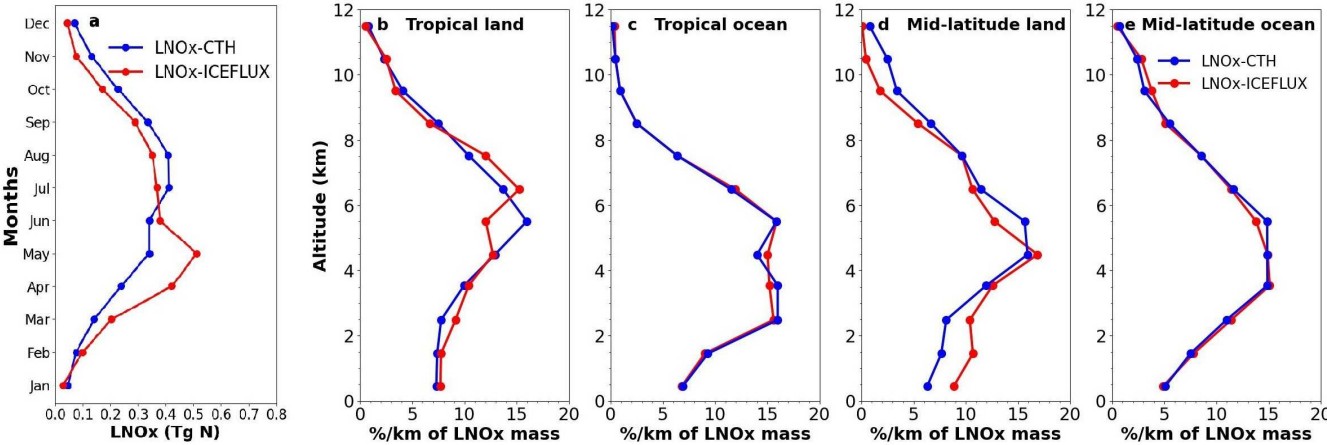

**Figure 3.** (a) Monthly LNOx emissions in Tg N from simulations LNOx-CTH and LNOx-ICEFLUX. (b–e) Vertical distribution of percentage of annual LNOx mass per kilometer from simulations LNOx-CTH and LNOx-ICEFLUX averaged over (b) tropical land, (c) tropical ocean, (d) mid-latitudinal land and (e) mid-latitudinal ocean.

due to strong downdraft, where the distribution is low and almost uniform up to 5–6 km as observed from the studies by Ott et al. (2010); Luhar et al. (2021). Nevertheless, the profile over mid-latitude lands from our study matches well with that from the study by Ott et al. (2010), with a maximum at 5 km. In our study, 13%–19% of total LNOx mass is estimated from surface to up to 2 km over tropical and mid-latitude lands and tropical oceans. LNOx production is suggested to be proportional to
atmospheric pressure by Goldenbaum and Dickerson (1993); Pickering et al. (1998). The vertical distribution of LNOx mass can be improved by replacing the simple distribution currently used, with the more detailed scheme developed by Pickering et al. (1998).

## 3.3 Vertical distribution of gases: effects of LNOx

### 3.3.1 Ozone

Numerous studies have demonstrated that LNOx emissions play a significant role in influencing the levels of ozone and other trace gases as a result of the oxidation of CO, $CH_4$ and volatile organic compounds (VOCs), particularly in the free troposphere (Luhar et al., 2021; Mao et al., 2021; Finney et al., 2016b; Liaskos et al., 2015). The changes in the annual mean of $O_3$ mixing ratio, due to the inclusion of LNOx in simulation LNOx-CTH, with respect to that from simulation noLNOx, averaged over four altitude bands (998–900, 900–750, 750–500 and 500–200 hPa) is presented in Figure 4. The changes in the same from
experiment LNOx-ICEFLUX with respect to LNOx-CTH, are also produced in Figure 5. The highest increase in simulated $O_3$ is observed in the altitude band 750–500 hPa, specifically over the tropics, by 6–10 ppbv. 4–6 ppbv increase over the tropics is also observed at the altitude bands 900–750 hPa and 500–200 hPa (mid- to upper troposphere). The increase in $O_3$ is maximum over the tropical region of America, central Africa, southern Asia and the Maritime continents in south-east Asia (Indonesia, Philippines and Malaysia). The above-mentioned regions with comparatively larger increase in $O_3$, identified for all the altitude

bands, are observed as regions with the largest convection depth and LNOx emissions (Banerjee et al., 2014). A higher value by 2–4 ppbv in annual mean $O_3$ is also observed from LNOx-ICEFLUX, with respect to LNOx-CTH over the tropical region for the altitude band 750–500 hPa, followed by 900–750 hPa band (Figure 5), even higher increase over the tropical region of America, central Africa and Tibetan Plateau is also found from LNOx-ICEFLUX. On the other side, the changes in $O_3$ mixing ratio are insignificant over mid-latitudes and polar regions from both the simulations with respect to noLNOx. The percentage

changes in the annual mean of $O_3$ mixing ratio from LNOx-CTH with respect to noLNOx, averaged over selected latitude and altitude bands present an overall improvement in tropospheric $O_3$ (Table S2 in supplementary). $O_3$ levels are significantly elevated by 10%–19% in the mid and upper troposphere (750–200 hPa), where $O_3$ production occurs efficiently (Dahlmann et al., 2011). Tropical mid- and upper tropospheres are more crucial in $O_3$ production as most lightning discharges occur in these regions (Luhar et al., 2024; Bucsela et al., 2019; Murray, 2016). A moderate (3%–5%) to low (1%–2%) increase in

annual mean of mid and upper tropospheric $O_3$ is also observed over mid-latitudes followed by the polar region. The increase is comparatively higher during late spring and early summer (May–August) being 6%–15% over mid-latitudes and 2%–4% over polar regions. Higher annual mean of $O_3$ mixing ratios are observed in LNOx-ICEFLUX compared to LNOx-CTH in the mid to upper troposphere across all latitude bands, likely due to comparatively higher LNOx production from LNOx-ICEFLUX in these regions.

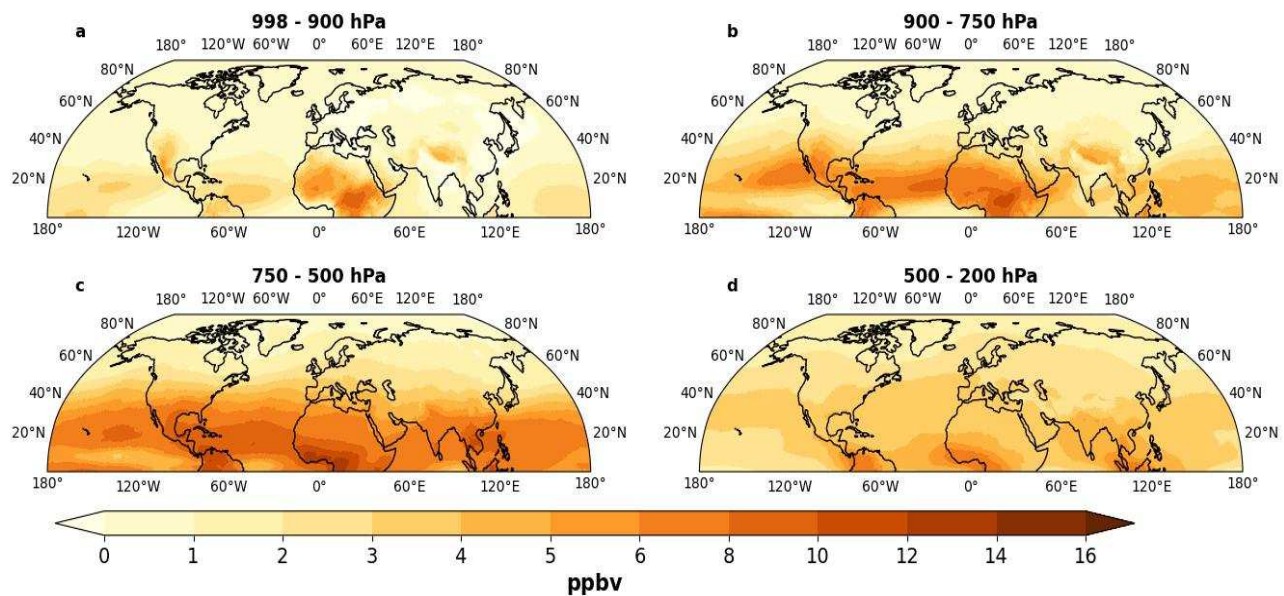

**Figure 4.** Changes in annual mean of $O_3$ mixing ratio from experiment LNOx-CTH w. r. t. noLNOx ($\Delta O_3$) at the altitude bands of (a) 998–900 hPa, (b) 900–750 hPa, (c) 750–500 hPa and (d) 500–200 hPa; positive and negative values represent the increase and decrease in the $O_3$ mixing ratio, respectively.

Figure 6 represents the vertical profile of annual mean $O_3$ mixing ratio from simulations and their comparison with the WOUDC ozone-sonde measurements, averaged for the stations over three latitude bands (0°–30° N, 30° N–60° N and 60°

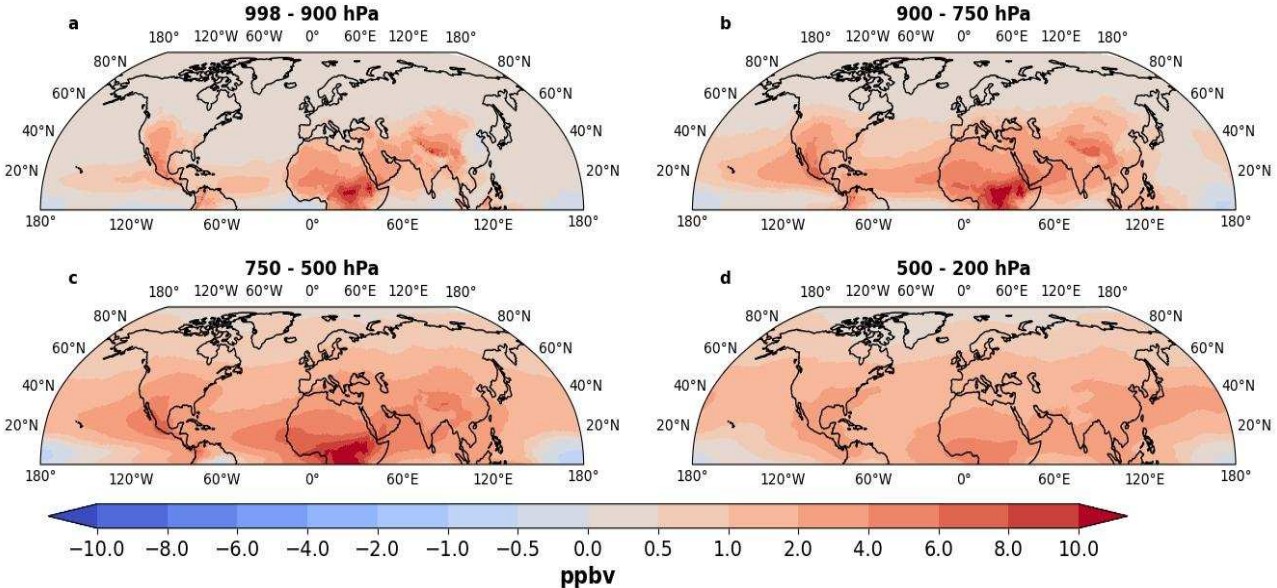

**Figure 5.** Changes in annual mean of $O_3$ mixing ratio from experiment LNOx-ICEFLUX w. r. t. LNOx-CTH at the altitude bands of (a) 998–900 hPa, (b) 900–750 hPa, (c) 750–500 hPa and (d) 500–200 hPa; positive and negative values represent the increase and decrease in the $O_3$ mixing ratio, respectively.

N–90° N). The upper tropospheric $O_3$ mixing ratio is moderately higher than that observed at surface at the tropics by 30%–60%, while it is 2–3 times higher over the mid-latitudes and polar region as observed from simulation LNOx-CTH (Table S2 in supplementary). The upper tropospheric $O_3$ over the polar region is also almost twice of that over the tropics (Table S2 in

supplementary). Notably, the vertical profile from observations represents an increasing $O_3$ mixing ratio with altitude, whereas, those from simulations show an overestimated $O_3$ mixing ratio near surface which tend to decrease near the boundary layer in the tropics (Figure 6). The simulated $O_3$ is also observed to be higher near surface and show a continuous increasing pattern over mid-latitudes and polar regions. It is visible in Figure 6, that the simulated $O_3$ mixing ratio from the experiments with LNOx, represents the measured $O_3$ adequately well, specifically in the free troposphere over tropics, where a large underestimation

is observed in simulated $O_3$ from noLNOx simulation. The absolute bias in simulated $O_3$ in the free troposphere, especially over the tropics, is reduced due to inclusion of LNOx in the model (Table S3 in supplementary). The bias is however lower for that from LNOx-CTH in comparison to LNOx-ICEFLUX. $O_3$ production efficiency due to LNOx is higher in the mid to upper troposphere, primarily because lower temperatures extend the lifetime of NOx, while enhanced photolysis rates further favour $O_3$ accumulation (Labrador et al., 2005). The high underestimation in the simulated $O_3$ mixing ratio in the altitude band

500–200 hPa, i.e., the upper troposphere and lower stratosphere, over mid-latitudes and polar regions still exists, even after the inclusion of LNOx, however lower underestimation is observed for LNOx-ICEFLUX. The underestimation suggests that the modeled stratosphere-troposphere exchange still requires significant refinement, and the cross-tropopause transport may not be adequately resolved due to the low model top.

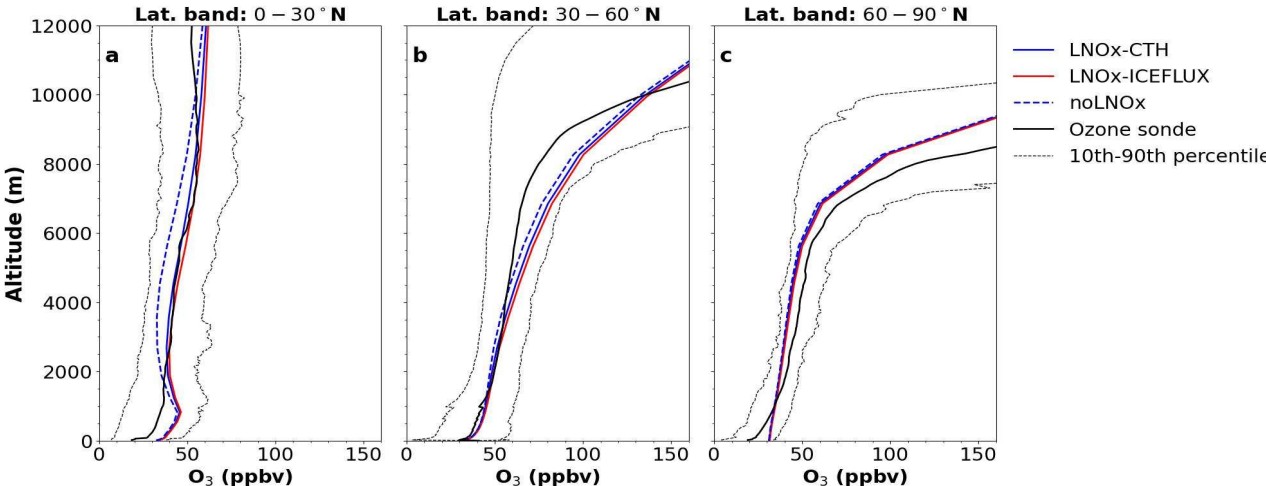

**Figure 6.** Vertical profile of annual mean of $O_3$ mixing ratio from noLNOx (blue dashed line), LNOx-CTH (blue solid line) and LNOx-ICEFLUX (red solid line) simulations and comparison with the WOUDC ozone-sonde measurements (black solid line), averaged for the stations over the latitude bands (a) $0°$–$30°$ N, (b) $30°$ N–$60°$ N and (c) $60°$ N–$90°$ N; the black dashed lines indicate the 10 and 90 percentiles of the WOUDC ozone-sonde measured values.

We also compare the vertical profiles of simulated $O_3$ with that from SHADOZ ozone-sonde measured data at stations of

Kuala Lampur, Hanoi, Costa Rica and Hilo, situated over the NH tropics. The plots are provided in supplementary material (Figure S1). Among the stations, at Kuala Lampur, the simulated $O_3$ profile from LNOx-ICEFLUX shows a good match with the observations at free troposphere, while the profile from LNOx-CTH aligns well with observations at Costa Rica. However the modelled $O_3$ profiles show under- and overestimation at most of the altitudes, respectively at Hanoi and Hilo, but overall replicate the observed altitudinal distribution quite well. The comparisons once again represents the effect of LNOx on $O_3$,

specifically at the free troposphere.

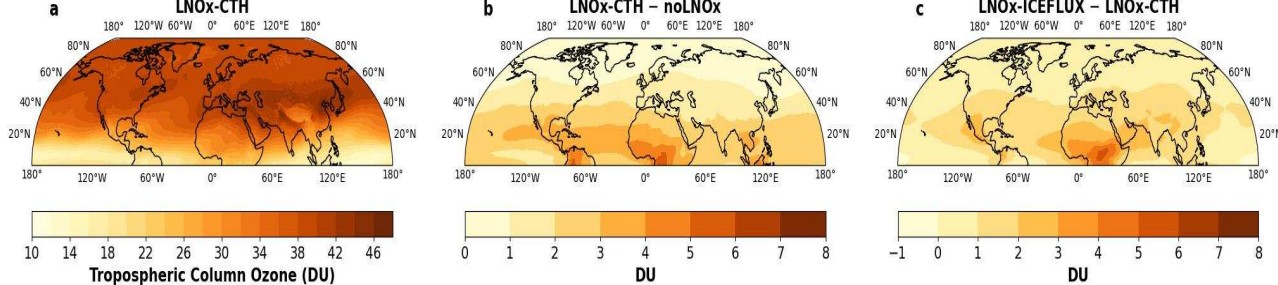

**Figure 7.** (a–c) Spatial distribution of (a) simulated tropospheric column of ozone (TCO) in DU, over the NH from experiment LNOx-CTH, (b) changes in simulated TCO from experiment LNOx-CTH w. r. t. noLNOx simulation, (c) differences in simulated TCO from experiment LNOx-ICEFLUX w. r. t. LNOx-CTH; positive and negative values represent the increase and decrease, respectively.

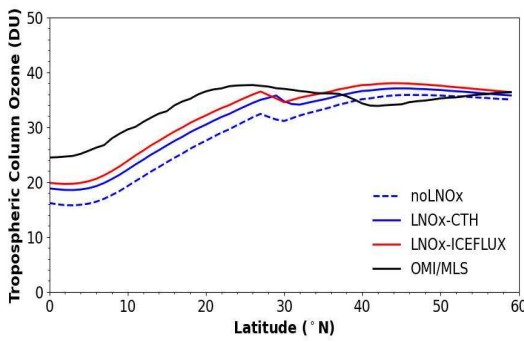

**Figure 8.** Zonal average of TCO from simulations over NH and their comparison with the same from OMI/MLS.

The monthly comparison of simulated $O_3$ from the LNOx-CTH and LNOx-ICEFLUX experiments with WOUDC ozone-sonde measurements, is presented in Figure S2 in the supplementary material, across two altitude (750–500 hPa and 500–200 hPa) and three latitude bands. The monthly variation reveals the highest peaks during March and October in the tropics, with the lowest levels observed during June–July for both altitude bands. A similar variation is noted at mid-latitudes and in the
polar regions for the 750–500 hPa altitude band, aligning well with the observed monthly trends. The simulated $O_3$ from both experiments closely matches the observed values at the tropics and mid-latitudes for both altitude bands, exhibiting a low bias of $\pm3$–10 ppbv. However, an overestimation of simulated $O_3$ is apparent during January–April, particularly over the tropics. While there is a good agreement between simulated and observed $O_3$ over the polar region for the 750–500 hPa altitude band, a significant underestimation of simulated $O_3$ is evident in the 500–200 hPa altitude band as discussed previously.

The spatial distribution of tropospheric column of ozone (TCO) from the LNOx-CTH simulation, along with the changes in TCO relative to the noLNOx and LNOx-ICEFLUX simulations, are shown in Figure 7. The TCO is observed to increase from the tropics toward higher latitudes, with the higher values occurring over the mid-latitudes (32–42 DU), particularly over Asian countries (40–44 DU; Figure 7a). The lower TCO values in the tropics are attributed to the model's top height being lower than the tropopause height in this region (refer to the Table S1 in supplementary material presenting the tropopause height).
The TCO from the LNOx-CTH simulation is higher by 2-4 DU at tropics, than that from the noLNOx simulation, while the LNOx-ICEFLUX simulation produces even higher TCO values compared to LNOx-CTH, especially over the tropics (Figures 7b–7c). The tropospheric $O_3$ burdens, estimated from the simulation LNOx-CTH and LNOx-ICEFLUX, are respectively 176 (150) Tg and 182 (155) Tg, over the NH and over the domain of 0°–60°N (presented inside parentheses). These burdens represent a 7%–11% increase relative to the noLNOx simulation (Table 8). Notably, the estimated $O_3$ burden in this study
aligns closely with observations from OMI/MLS (159 Tg) for the domain of 0°–60°N. The spatial distribution of TCO from OMI/MLS is shown in Figure S3 in supplementary material. A comparison of the zonal mean TCO over 0°–60°N reveals good agreement between the simulated TCO and OMI/MLS observations in the mid-latitudes (Figure 8). However, the simulations underestimate TCO in the tropics by 7%–26%, owing to the limited model top height, which excludes part of the troposphere above it. Despite this limitation, incorporating LNOx into the model leads to significant improvements in simulated TCO.

## 3.3.2 NO$_2$

Figure 9a represents the spatial distribution of NO$_2$ column density estimated from LNOx-CTH. A high NO$_2$ column density of 2–3 $\times$ 10$^{15}$ molecules cm$^{-2}$ is observed over the southern and eastern Asia (India and eastern China), north-west Europe and eastern part of USA. The spatial variation in NO$_2$ column matches well with that obtained from OMI observations (Figure S4 in supplementary material), highlighting elevated NO$_2$ column densities in countries with significant industrial activities (Cooper et al., 2022). A decrease in NO$_2$ column density (0.2–0.6 $\times$ 10$^{15}$ molecules cm$^{-2}$) due to inclusion of LNOx emissions, is primarily observed over the above mentioned regions with high NO$_2$ pollution (Figure 9b). The inclusion of LNOx in model increases large-scale O$_3$ and OH concentrations, therefore reducing the lifetime of NOx through oxidation reactions with HOx including OH (Labrador et al., 2005; Schumann and Huntrieser, 2007). Figure S5 in supplementary material depicts the increase in HNO$_3$ column density over the above mentioned region, supporting the fact that NO$_2$ is oxidized and converted to the HNO$_3$, increasing the column density of HNO$_3$. Hence, rapid conversion of NO$_2$ into other compounds, such as HNO$_3$, leads to its subsequent removal and a net decrease in NO$_2$ column density over the regions with high anthropogenic pollution. The Figure S6 in supplementary material, showing changes in annual mean NO$_2$ mixing ratio (in ppbv) from experiment LNOx-CTH with respect to noLNOx, demonstrates a decrease in NO$_2$ by 0.1–0.3 ppbv over the regions with higher anthropogenic NO$_2$ pollution as mentioned above, at the altitude band 998–900 hPa, i.e., mostly near surface followed by the altitude band 900–750 hPa. A very small increase (0.05 ppbv) is observed over most part of NH at the higher altitude bands (750–500 and 500–200 hPa), due to inclusion of LNOx emissions. Overall the NO$_2$ column density decreases over the regions with high anthropogenic pollution. Again, small increases of 0.1–0.3 $\times$ 10$^{15}$ molecules cm$^{-2}$ are observed over Africa, South America, south-east Asia, the Maritime Continent and the tropical oceans, where NO$_2$ pollution is relatively lower (Figures 9a and 9b). Notably, NO$_2$ column densities from LNOx-ICEFLUX are higher by 0.6–0.8 $\times$ 10$^{15}$ molecules cm$^{-2}$ compared to LNOx-CTH over southern Asia, central Africa, and parts of the United States (Figure 9c).

The zonally averaged NO$_2$ column distribution (Figure 10) reveals elevated column densities over the tropics, especially between 20°–30°N, and the mid-latitudes, even in the absence of LNOx emissions. The zonal averages range from 0.35–1.75 $\times$ 10$^{15}$ molecules cm$^{-2}$ in these regions, which is nearly double the values observed at higher latitudes (60°–90°N). The peak at 20°–30°N, of 1.75 $\times$ 10$^{15}$ molecules cm$^{-2}$, is due to the high NO$_2$ column density estimated from simulations over the southern and south-east Asia due to high NO$_2$ emissions from larger industrial activities. This peak is however not observed in OMI observations. On the other hand, a study by Luhar et al. (2021) has depicted that the NO$_2$ column density obtained from Copernicus Atmosphere Monitoring Service (CAMS) reanalysis data, shows a peak of 1.5 $\times$ 10$^{15}$ molecules cm$^{-2}$ at this latitude band (20°–30°N), where OMI underestimates the NO$_2$ column density. The higher uncertainty in OMI retrieved NO$_2$ columns, as compared with available satellite observations (GOME-2, SCIAMACHY and TROPOMI) is considerable in this regards. The uncertainties are primarily due to instrumental errors, limitations of the OMI sensor in capturing the NO$_2$ below the cloud level, vertical profile assumptions and surface reflectivity (Bucsela et al., 2013; Boersma et al., 2018). A secondary maximum in NO$_2$ column density is identified between 35°–45°N from simulations as well as from satellite observations. However, the simulated NO$_2$ column density is underestimated at mid-latitudes by 20%–40%. At

higher latitudes (60°–90°N), where the magnitudes are comparatively lower, the simulated NO$_2$ column density matches well
with satellite-based observations. Overall, the zonally averaged NO$_2$ column densities from the simulations closely replicate
satellite observations, except for a pronounced peak at 20°–30°N from simulated NO$_2$. The tropospheric burden of NO$_2$ is 146
Gg from LNOx-CTH, being comparable to that from noLNOx and 11% lower than that estimated from OMI (Table 8). The
burden estimated from LNOx-ICEFLUX is 3% higher and 8% lower than the LNOx-CTH and OMI, respectively.

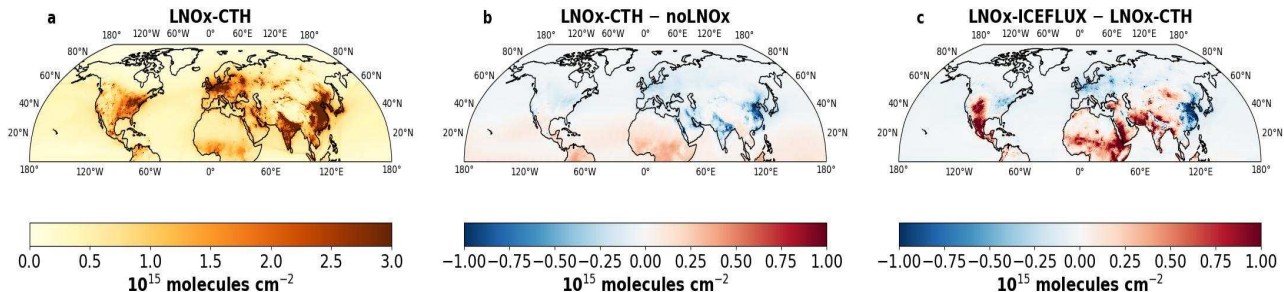

**Figure 9.** (a–c) Spatial distribution of (a) simulated NO$_2$ column density in $10^{15} \times$ molecules cm$^{-2}$, over the NH from experiment LNOx-CTH, (b) changes in simulated NO$_2$ column density from experiment LNOx-CTH w. r. t. noLNOx simulation, (c) differences in simulated NO$_2$ column density from experiment LNOx-ICEFLUX w. r. t. LNOx-CTH; positive and negative values represent the increase and decrease, respectively.

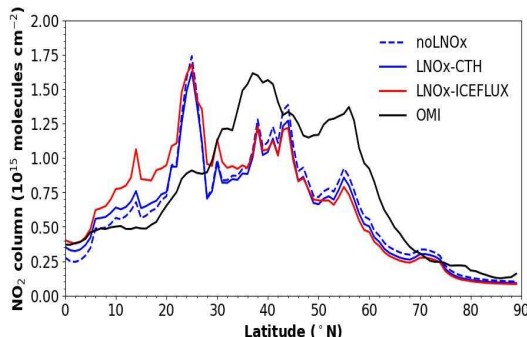

**Figure 10.** Zonal average of NO$_2$ column density from simulations over NH and their comparison with the same from OMI.

## 3.4   Impacts of LNOx on surface-level O$_3$ and NO$_2$

The effect of LNOx on surface level O$_3$ and NO$_2$ over NH is analysed in this section. The spatial distribution of the annual
mean of O$_3$ mixing ratio and NO$_2$ at the surface from experiment LNOx-CTH and changes in the mixing ratio due to inclusion
of LNOx ($\Delta$O$_3$ and $\Delta$NO$_2$), are presented in Figures 11a, 11d and 11b, 11e, respectively. Other than natural sources (e.g.,
lightning, soil-NOx emissions), emissions from fossil fuel combustion for transportation, industrial activities, energy generation
and biomass burning also have a profound influence on tropospheric O$_3$ and NO$_2$ concentrations (Lelieveld and Dentener,
2000; van der A et al., 2008; Butler et al., 2020). In our study, the O$_3$ mixing ratio at the surface varies spatially with higher

values ranging between 35–45 ppbv over the latitude band of $10°–50°N$, specifically over the lands, being almost 1.5–2 times of that observed over the rest of the NH. The $NO_2$ at the surface, is within a range of 0.5–2 ppbv over most parts of the NH showing higher magnitudes over USA, western Europe, India, eastern China and Japan (5–10 ppbv). To indicate the impact of LNOx emissions on changes in surface-level $O_3$ and $NO_2$, annual mean of mixing ratio obtained from experiment LNOx–CTH is compared with respect to that from noLNOx. The positive and negative values of $\Delta O_3$ and $\Delta NO_2$ depict an increase and decrease, respectively, in surface mixing ratio (Figures 11b and 11e). The study shows an overall increase in surface $O_3$ by 1–3 ppbv over most parts of the tropical lands and mid-latitudes up to $50°N$ (Figure 11b), while the increase is almost negligible over $50°–90°N$ ($<1$ ppbv). A comparatively larger increase by 3–5 ppbv is observed over tropical parts of America and Africa and the Tibetan Plateau, but is particularly noteworthy (5–10 ppbv) over the central part of Africa, which is a hotspot location with high lightning flash rate (refer Section 3.1). The $O_3$ level from LNOx-ICEFLUX is even higher (2–4 ppbv) than that estimated from LNOx-CTH (Figure 11c). Unlike $O_3$, $NO_2$ exhibits both increase and decrease in mixing ratio at the surface as an effect of lightning (Figure 11e), but by a lesser magnitude (0.01–0.1 ppbv). While the increase is observed over South America, Africa, the Maritime Continent and south-east Asia, a decrease in $NO_2$ mixing ratio is also there over India, eastern and south-west Asia and most of the continents north of $30°N$. $O_3$ and $NO_2$, both exhibit a slight increase over the Atlantic and Pacific oceans in the tropics. The magnitudes and spatial patterns of $\Delta O_3$ and $\Delta NO_2$ from our study bear a resemblance to those from recent studies (Murray, 2016; Li et al., 2022; Cheng et al., 2024). The increase and decrease in $NO_2$ surface mixing ratio from LNOx-ICEFLUX in respect to LNOx-CTH is represented in Figure 11f. The impact on surface $O_3$ and $NO_2$ concentrations is a localized effect of thunderstorms, crucially influenced by the specific photochemical conditions in the area (Murray, 2016). An increase in surface $O_3$ levels due to LNOx suggests the NOx concentration to be below the titration threshold (Pawar et al., 2012b).

The statistical analyses are also done comparing the simulated mixing ratios at surface to the observations and presented in Table 6. The agreement between simulated and observed $O_3$ is considered good, as indicated by lower values of RMSE (10.7–11 ppbv), MAB (6.5–7.1 ppbv), and NME (26.9%–27.8%). In contrast, the comparison for simulated $NO_2$ with observations shows higher NME (51.5%–52.7%). Figures S7(a–d) in supplementary material represent the absolute bias in the simulated annual mean of $O_3$ and $NO_2$ at the surface from experiment LNOx-CTH and LNOx-ICEFLUX, compared to the observations at available stations over the NH. The simulated $O_3$ and $NO_2$ mixing ratio is close enough to the observations at most of the stations over Europe and China, and also over the USA for $O_3$ and Canada and south America for $NO_2$. Higher bias for $O_3$ is, nonetheless, observed at stations over Canada, South America and few stations over eastern China. However, the inclusion of LNOx does not significantly impact the statistical scores. A detailed analysis of the altitude-wise changes in the mixing ratio of $O_3$, due to the impact of LNOx, is therefore necessary and has already been discussed in the Sections 3.3.1 and 3.3.2.

### 3.5 Impacts on tropospheric OH burden and CH$_4$ lifetime

We also have evaluated the effects of LNOx on tropospheric chemistry in terms of changes in the burden of a major oxidant (OH) and the lifetime of trace gas $CH_4$. Table 7 illustrates the concentration of OH from experiment LNOx-CTH and the changes in concentration of OH with respect to that from noLNOx averaged over selected latitude and altitude bands. The OH

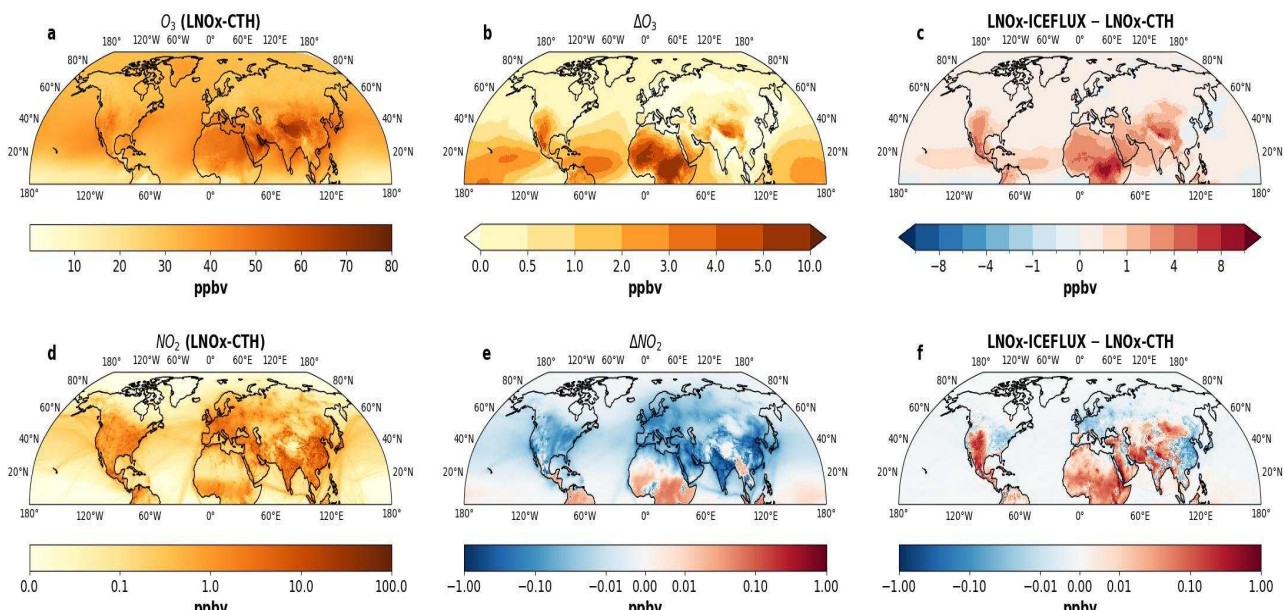

**Figure 11.** (a, d) Spatial distribution of annual mean mixing ratio in ppbv at the surface over NH from experiment LNOx-CTH for (a) $O_3$, (d) $NO_2$; (b, e) changes in mixing ratio at the surface due to inclusion of LNOx emissions (LNOx-CTH − noLNOx) for (b) $O_3$, (e) $NO_2$; (c, f) differences in mixing ratio at the surface from experiment LNOx-ICEFLUX w. r. t. LNOx-CTH for (c) $O_3$, (f) $NO_2$; positive and negative values show an increase and decrease, respectively.

**Table 6.** Statistical analysis comparing simulated mixing ratio at surface to the ground-based observations for $O_3$ and $NO_2$. The positive and negative values of MAB represent that the simulated surface contration is higher and lower than the observation, respectively.

| | Mean observed surface conc. over NH (ppbv) | Total number of stations | noLNOx | | | LNOx-CTH | | | LNOx-ICEFLUX | | |
|---|---|---|---|---|---|---|---|---|---|---|---|
| | | | RMSE (ppbv) | MAB (ppbv) | NME (%) | RMSE (ppbv) | MAB (ppbv) | NME (%) | RMSE (ppbv) | MAB (ppbv) | NME (%) |
| $O_3$ | 32.48 | 5185 | 10.9 | 6.5 | 27.3 | 10.7 | 6.5 | 26.9 | 11.0 | 7.1 | 27.8 |
| $NO_2$ | 14.3 | 3857 | 9.2 | 6.4 | 51.5 | 9.3 | 6.6 | 52.7 | 9.3 | 6.6 | 52.7 |

concentration from the LNOx-ICEFLUX is also compared with that from LNOx-CTH in Table 7. The OH concentration in our study, over the tropics, is almost twice and 6–7 times higher than that over mid-latitudes and polar regions, respectively, which is consistent with the study by Mao et al. (2021). Again these concentrations are close enough to those values obtained in a multi-model study by Naik et al. (2013), but shows a higher OH concentration at tropical mid-troposphere ($22–26 \times 10^5$ molecules $cm^{-3}$), unlike the studies by Naik et al. (2013); Luhar et al. (2021). Higher OH concentration in the upper troposphere is reported by Banerjee et al. (2014), due to transportation of water vapour through convection to the upper troposphere promoting the OH production due to reaction of exited state oxygen with water vapour. The annual mean OH concentration over NH, from our study, is $14.5 \times 10^5$ and $15.4 \times 10^5$ molecules $cm^{-3}$ from LNOx-CTH and LNOx-ICEFLUX, respectively, being 6.6% and 13.2% higher than that obtained from noLNOx simulation (Table 8). The annual mean OH concentration is higher by 30%–38% in comparison to the multi-model mean obtained from ACCMIP simulations ($11.1 \pm 1.8 \times 10^5$ molecules $cm^{-3}$; Naik et al., 2013; Voulgarakis et al., 2013). We find an increase in OH concentration due to LNOx, which is again the largest

over free troposphere at tropics (11%–28%), followed by mid-latitudes (Table 7). A 5%–7% higher OH concentration is also observed from LNOx-ICEFLUX in comparison to that from LNOx-CTH at the free troposphere. A warmer atmosphere at tropics and high humidity favour the increase in OH and a faster OH to $CH_4$ reaction, causing a shorter $CH_4$ lifetime at these regions (Voulgarakis et al., 2013). The geographical distribution of changes in OH due to lightning is usually affected by the lightning parameterization used (Gordillo-Vázquez et al., 2019). The spatial distribution of changes in simulated OH concentration from LNOx-CTH w. r. t. noLNOx and from LNOx-ICEFLUX w. r. t. LNOx-CTH, at selected altitude bands, are also presented, respectively in Figures S8 and S9 in the supplementary material. The OH burden over NH is increased by 14% and 24% in LNOx-CTH and LNOx-ICEFLUX, respectively, from 0.082 Gg, estimated in simulation noLNOx (Table 8).

In our study, we have estimated $CH_4$ lifetime ($\tau_{CH_4}$) due to chemical loss, mainly due to reaction with OH. The average lifetime over NH are 4.84 and 4.5 years, as obtained respectively from experiments LNOx-CTH and LNOx-ICEFLUX, which are reduced respectively by 11% and 17% compared to the estimate from the noLNOx simulation (Table 8). The $CH_4$ lifetime as estimated from previous modelling studies, is within a range of 7–14 years (Naik et al., 2013; Lelieveld et al., 2016) and is visibly underestimated in our study. The $CH_4$ concentration is considered from chemical boundary conditions from CAMS reanalysis dataset of atmospheric compositions produced by ECMWF, as in our study, $CH_4$ anthropogenic emissions are not taken into account. The annual mean $CH_4$ burden (1930–1933 Tg) estimated in this study over the NH is ≈20% lower than the multi-model mean $CH_4$ burden, obtained from ACCMIP simulations (Naik et al., 2013), considering half of the global $CH_4$ burden over the NH (≈2406 Tg). As mentioned above, the OH concentration is also overestimated in our study. Therefore, the lower $CH_4$ burden and higher chemical loss due to reaction with OH, cause the underestimated lifetime of $CH_4$ in this study, even in the absence of LNOx. Therefore, the underestimated $CH_4$ lifetime in the present study is not attributed to LNOx but likely stems from other factors including issues related to deficiencies in $CH_4$ burden, the chemistry or photolysis schemes. Addressing and resolving these concerns will require further investigation in future studies. The $CH_4$ lifetime is underestimated especially over tropics showing values of 2–4 years (Figure S10 in supplementary material). The lifetime increases with higher latitudes and is maximum at polar region (40–60 years), which matches well with the estimated values from the study by Lelieveld et al. (2016), over higher latitudes (45°–90°N).

**Table 7.** Analysis of simulated OH concentration averaged over selected latitude and altitude bands. ΔOH represents changes in OH from experiment 'LNOx-CTH' w. r. t. that from experiment 'noLNOx'; positive and negative values represent the increase and decrease in OH concentration, respectively.

| Latitude band | 0°–30° N | 30° N–60° N | 60° N–90° N | 0°–30° N | 30° N–60° N | 60° N–90° N | 0°–30° N | 30° N–60° N | 60° N–90° N |
|---|---|---|---|---|---|---|---|---|---|
| Altitude band (hPa) | OH concentration ($10^5$ molecules $cm^{-3}$) from LNOx-CTH | | | ΔOH (%) | | | LNOx-ICEFLUX − LNOx-CTH (%) | | |
| 500–200 | 15.1 | 7.3 | 2.6 | 20.8 | 7.3 | −3.7 | 5.2 | 2.6 | 1.5 |
| 750–500 | 26.0 | 11.3 | 3.1 | 28.7 | 8.6 | −6.1 | 7.6 | 3.5 | 1.7 |
| 900–750 | 22.4 | 11.9 | 3.2 | 11.4 | 0 | −3.0 | 5.8 | 2.6 | 1.1 |
| 998–900 | 19.5 | 13.4 | 2.8 | 2.1 | −2.9 | −3.7 | 4.2 | 1.9 | 0.8 |

**Table 8.** Tropospheric $O_3$, $NO_2$, OH burden and $CH_4$ lifetime from simulation experiments. The numbers within parentheses represent the tropospheric $O_3$ burden over the domain from $0°$–$60°N$.

|  | noLNOx | LNOx-CTH | LNOx-ICEFLUX |
|---|---|---|---|
| $O_3$ burden (Tg)[†] | 164 (138) | 176 (150) | 182 (155) |
| $NO_2$ burden (Gg)[††] | 146 | 146 | 150 |
| OH concentration (molecules $cm^{-3}$) | $13.6 \times 10^5$ | $14.5 \times 10^5$ | $15.4 \times 10^5$ |
| OH burden (Gg) | 0.082 | 0.094 | 0.102 |
| $CH_4$ lifetime due to chemical loss (yr) | 5.45 | 4.84 | 4.5 |

[†]Tropospheric $O_3$ burden from OMI/MLS, estimated over the domain $0°$–$60°N$, is 159 Tg.

[††]Tropospheric $NO_2$ burden over NH from OMI is 164 Gg.

## Conclusion

This study evaluates the incorporation of lightning-produced NOx (LNOx) into the CHIMERE chemistry-transport model to assess its impact on tropospheric ozone ($O_3$) over the Northern Hemisphere (NH). A classical lightning parameterization based on cloud top height (CTH), developed by Price and Rind (1992), is applied (experiment: LNOx-CTH) with modifications to better align modelled flash rates over lands and oceans to satellite observations. Additionally, flash rates are computed using an updated ice flux based lightning scheme (experiment: LNOx-ICEFLUX; Finney et al., 2014). We perform a detailed evaluation of model simulations, integrating in situ measurements and satellite observations to critically assess the reliability and applicability of these parameterizations. The annual flash frequencies over the NH are 20.7 and 21.6 flashes $s^{-1}$ as estimated from the LNOx-CTH and LNOx-ICEFLUX experiments, respectively. For LNOx-ICEFLUX, a correction factor of 5 is applied to the simulated annual flash frequencies. The estimated LNOx emissions are 2.8 and 3.1 Tg N $yr^{-1}$ from LNOx-CTH and LNOx-ICEFLUX experiments, respectively. Our study provides a comparative assessment of these two lightning parameterizations, evaluating their influence on modelled lightning flashes, LNOx emissions and tropospheric distribution of $O_3$ and trace gases, with implications for improving both the parameterizations and the model.

The major outcomes from our study are delineated here. Annual flash rates in tropical land and NH regions from both experiments, as well as in mid-latitudes from LNOx-ICEFLUX, show good agreement with satellite observations. Both the ICEFLUX and CTH schemes as implemented in CHIMERE, reproduce the seasonal cycle of lightning flash rates correctly over the lands. LNOx emission peak during May–August, contributing 60%–70% of the total annual emissions, with most emissions concentrated in the tropics and mid-latitudes, and 60%–65% occurring in the mid-tropospheric region. The inclusion of LNOx emissions in CHIMERE significantly improves the simulated tropospheric $O_3$ distribution, particularly in the free troposphere over the tropics. A significant bias at the upper troposphere and lower stratosphere at higher latitudes, however highlights the necessity of improving the representation of stratosphere-troposphere exchange processes in the model. The model adequately simulates the $O_3$ and $NO_2$ burden over the NH compared to satellite observations, showing a 7%–11% increase in $O_3$ burden from 164 Tg, due to the inclusion LNOx. Our study demonstrates that the inclusion of LNOx consequently reduces the overall $NO_2$ column density over regions with high anthropogenic pollution. Additionally, it leads to a 14%–24% increase in OH

burden from 0.082 Gg and a 11%–17% reduction in $CH_4$ lifetime compared to the without LNOx scenario, though there remains scope for refining OH-related chemistry.

Our study underscores that, despite its simple representation, the CTH scheme better captures the spatial variability of flashes compared to satellite observations, outperforming the ICEFLUX scheme. However, the limitations of the CTH scheme in capturing mid-latitude flashes highlight the efficacy of the ICEFLUX scheme in these areas. Additionally, improving convective parameterization is crucial for better representation of oceanic flash rates. The ICEFLUX scheme also faces challenges in accurately simulating high-energy, less frequent flashes over oceans, emphasizing the need to incorporate additional factors alongside ice flux. The challenges to constrain parameters, such as cloud ice content and updraft mass flux, which are utilized in flux-based lightning schemes, continue due to limited available observations. A recent study by Cummings et al. (2024), which evaluated eighteen lightning parameterization schemes, demonstrates that those based on storm kinematics and structure performed better than the microphysical schemes, in their study. Developing integrated parameterizations that incorporate both storm kinematics and microphysical processes, along with improved observational constraints, may provide a more robust and accurate representation of lightning flash rates, particularly in complex storm environments.

*Code and data availability.* The CHIMERE model (v2023r2) is available on the website at https://www.lmd.polytechnique.fr/chimere/. The in situ measurements and satellite data used in the study are all freely downloadable from cited URLs.

*Author contributions.* SG analysed the model output and data, downloaded satellite data, made plots, conceptualised and prepared the first version of the manuscript. AC performed the simulations and contributed to the measurement data collection. All co-authors have participated in conceptualisation of the study, the interpretation and discussion of the results, and the drafting of the final manuscript.

*Competing interests.* The authors declare that they have no conflict of interest.

*Acknowledgements.* This study was supported by the Agence de l'Environnement et de la Maîtrise de l'Energie (ADEME) within the framework of the ESCALAIR project (grant: 2162D0017). We also acknowledge the use of computational resources provided by the GENCI GEN10274 project.

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
