# Peer review of "Representing improved tropospheric ozone distribution over the Northern Hemisphere by including lightning NOx emissions in CHIMERE"

_EGUsphere, 2024_

## Author Response (AR1)

**Review replies for the manuscript entitled "Representing improved tropospheric ozone distribution over the Northern Hemisphere by including lightning NOx emissions in CHIMERE" (egusphere-2024-3087) by Sanhita Ghosh et al.**

We sincerely thank the editor and both the reviewers for their constructive feedback and insightful comments on our manuscript. Below, we address each comment in detail and outline the corresponding revisions made to improve the manuscript.

**Reviewer 1**

**Summary**

The authors implement a cloud-height based flash rate scheme into the regional model CHIMERE and then evaluate the impact of lightning-NOx production on ambient ozone and $NO_2$ concentrations, ozone profiles, the tropospheric OH burden, and the methane lifetime. The magnitude of the lightning-NOx source is biased high, which has implications for the LNOx increment, OH burden, and CH4 lifetime. Thus, I think the authors need to re-run the simulation with a lower LNOx source, and then spend more time evaluating the lightning parameterization. As the LNOx scheme is not new, it would help if they modify it to better simulate the seasonal cycle. Thus, I would suggest rejecting the manuscript – at least for now.

**Reply:** We thank the reviewer for the valuable comments. As per the suggestions by the reviewer, we have re-run the simulation with parameterization based on cloud top height (CTH), applying modifications to better align the modelled flash rates over lands and oceans, to the satellite observations. Additionally, flash rates are computed using an updated ice flux based lightning scheme (ICEFLUX) for a comparative study. We perform a detailed evaluation of modelled lightning flashes, simulated ozone ($O_3$) and trace gases, integrating in situ measurements and satellite observations, to critically assess the reliability and applicability of these parameterizations. Simulations with improved model configuration for advection, capture the seasonality in lightning flashes as observed from satellite-based measurements. We have addressed and resolved most of the issues identified by the reviewer. The magnitude and spatial distribution of modelled flashes are now improved and the impacts due to inclusion of lightning-induced NOx emissions (LNOx) on the tropospheric distribution of $O_3$ and nitrogen dioxide ($NO_2$) are thoroughly studied. Vertical profile of $O_3$ has been evaluated by comparing with measured values obtained from WOUDC ozone-sonde as well as SHADOZ ozone-sonde. The column densities of $O_3$ and $NO_2$ have also been analysed in comparison to the satellite observations. We also have represented the effects of LNOx on the hydroxyl radicals (OH) burden, which further impact the atmospheric lifetime of trace gas methane ($CH_4$).

The inclusion of LNOx in chemistry-transport models has been a research topic for several decades. In this study, we build on previous work by implementing the ICEFLUX scheme in the chemistry-transport model CHIMERE and comparing it with the CTH scheme. Further, validation and evaluation of lightning parameterizations across different models (global and mesoscale) are essential for fully assessing their robustness and applicability, highlighting the importance of this study. Therefore, with the new improved simulations and detailed evaluation of the results, we believe that the scientific values of the manuscript has been increased to meet the standard of the journal 'Atmospheric Chemistry and Physics'.

Point by point replies to the reviewer's comments are provided below.

**Main Comments:**

*The evaluation of the flash rate scheme needs to be enhanced.*

1. L8 L202: 8.82 Tg N yr$^{-1}$ is too high of a production rate for just the Northern Hemisphere. It is consistent with $\approx$15 Tg N yr$^{-1}$ for the globe, a value that is well outside of the typically cited 2–8 Tg N yr$^{-1}$ range. You need to justify why you chose that value or choose a lower value.

**Reply:** We agree with the reviewer that 8.82 Tg N yr$^{-1}$ is a high production rate of LNOx emissions for the Northern Hemisphere (NH), as the LNOx emissions typically range within 2–8 Tg N yr$^{-1}$ as obtained from previous modelling studies.

We have now re-run the simulation with cloud top height (CTH) scheme, with modification in the equations for estimating flash rates over lands and oceans, to reconcile the estimated flash rates with the satellite-based observations. Additionally, we have adopted values for NO molecule production per flash by considering flash energies of 3 GJ for cloud-to-ground (CG) flashes and 0.9 GJ for in-cloud (IC) flashes (Schumann and Huntrieser, 2007). The NO production rate is $14.2 \times 10^{16}$ molecules NO J$^{-1}$. Using these updated parameters, we obtained an average of 332 moles of NO per flash with a mean IC to CG flash ratio of 3.09. With these new values, the estimated NOx emissions from lightning (LNOx) over the NH amount to 2.8 Tg N yr$^{-1}$, which falls within the range of 2–8 Tg N yr$^{-1}$ reported in earlier studies (Schumann and Huntrieser, 2007; Finney et al., 2016; Nault et al., 2017).

Additionally, we carried out simulations with a lightning parameterization based on the ice flux (ICEFLUX) for a comparative study. The LNOx emission estimated using ICEFLUX scheme is 3.1 Tg N yr$^{-1}$ over the NH, after applying a correction factor of 5 to the simulated flash rates. The LNOx emissions from our study align well with the estimates by Luhar et al. (2021), which ranged from 2.39 to 3.41 Tg N yr$^{-1}$ over the NH, respectively using CTH and a new parameterization developed by Luhar et al. (2021). These values of LNOx emissions have been incorporated into our study to further assess their impacts on ozone and other trace gases.

**Changes in the manuscript:** Please see the lines 364–365, "The estimated annual NO emissions from lightning are 2.8 Tg N yr$^{-1}$ and 3.1 Tg N yr$^{-1}$ over the NH, respectively from LNOx-CTH and LNOx-ICEFLUX simulations (Table 5)."

Also refer the Subsections 2.2.1 and 2.2.2 in the manuscript for the details of CTH and ICEFLUX schemes, respectively and Section 2.3 for the estimation of LNOx emissions.

Lines 182–185, "In this study, we adopt flash energies of 3 GJ for CG flashes and 0.9 GJ for IC flashes, along with a NO production rate of $14.2 \times 10^{16}$ molecules NO J$^{-1}$ (Schumann and Huntrieser, 2007). Using these values, we calculate the NO production in moles per flash as described in Equation 9, yielding a mean value of 332 moles of NO per flash. This estimation considers CG flashes as 25% of the total lightning flashes."

$$P(\text{CG, NO}) = 697.44\,\text{moles flash}^{-1},$$
$$P(\text{IC, NO}) = 199.27\,\text{moles flash}^{-1}$$

2. L119: I don't understand why the flash rate you obtain is so high given that you've scaled it to match a satellite product. What satellite data are you comparing to? What is the estimated flash rate of the satellite-based observation? What is the Northern Hemisphere flash rate after applying the scaling factor?

**Reply:** As mentioned in the previous reply, we have now re-run the simulation with CTH scheme, applying modifications in the equations for estimating flash rates over lands and oceans, to reconcile the estimated flash rates with the satellite observations.

We also have carried out simulations with an ice flux based lightning parameterization (ICEFLUX). The flash rates obtained from Lightning Imaging Sensor on the International Space Station platform (ISS-LIS, domain: $\pm 55°$ latitudes) for the year 2018 has been used for the comparison with simulated flash rates. Additionally, we also have flash rates estimated from the combined climatology product of satellite observations from the Optical Transient Detector (OTD) and the Lightning Imaging Sensor (LIS) observation (LIS/OTD) for the period May, 1995 to December, 2014.

The annual flash frequencies over the NH is estimated as 20.7 and 21.6 flashes $s^{-1}$, respectively from the simulations with CTH and ICEFLUX schemes, are in alignment to the satellite observations. The flash frequencies are 23.6 and 26.4 flashes $s^{-1}$, respectively from ISS-LIS and LIS/OTD over NH. The flash frequency estimated with ICEFLUX scheme, is divided by a factor 5 to reconcile with the satellite-observed frequencies.

**Changes in the manuscript:** Please see the lines 339–344 in manuscript, "The annual flash frequencies over the NH is estimated as 20.7 and 21.6 flashes $s^{-1}$, respectively from the LNOx-CTH and LNOx-ICEFLUX experiments (Table 5). These values are consistent with the satellite observations (23.6 and 26.4 flashes $s^{-1}$, respectively from ISS-LIS and OTD/LIS observation over NH) as well as to that obtained in recent model-based studies over NH (Luhar et al., 2021). While no scaling factor is applied in the estimated flash rates from LNOx-CTH simulation, the flash frequency from LNOx-ICEFLUX is divided by a factor 5 to reconcile with the satellite-observed frequency. Accordingly, the evaluation of the results from LNOx-ICEFLUX has been conducted using flash rates adjusted by this factor."

3. L210: You note that the model does not capture the seasonality of flash rates. Could you compare the monthly model flashes for 0 to 35 N with OTD/LIS climatology. It would also be interesting to compare the flash rates with the diel OTD/LIS climatology. Is this a problem with the model's cloud top heights or with the scheme itself?

**Reply:** As mentioned previously, we have re-run the simulations. Currently, with improved configurations for advection in the model, we have conducted the simulations (for the year 2018) to estimate lightning flashes with CTH and ICEFLUX schemes. We have evaluated monthly flash rates from both the simulations and compared to the satellite observed flash rates from ISS-LIS (for the year 2018) and LIS/OTD (monthly climatology data for the priod of May, 1995 to December, 2014).

Flash rates from the LNOx-CTH experiment show a clear seasonal cycle, with peaks occurring in May–August over NH and lands. During the winter months (November–January), flash rates drop significantly, being 5–7 times lower than the summer (May–August) peak values. The seasonal variation observed in the LNOx-ICEFLUX experiment and satellite observations closely aligns with this trend. The modelled monthly averaged flash rates from the simulations exhibit a strong positive correlation with satellite observations (ISS-LIS and OTD/LIS), with correlation coefficients ranging from 0.85 to 0.97. This agreement is consistent across all latitude bands over the NH and land regions from both the experiments (LNOx-CTH and LNOx-ICEFLUX). These findings indicate that the simulations successfully capture the seasonal variability of flash rates over the NH and lands.

**Changes in the manuscript:** Please see the lines 314–331 in manuscript, "Figure 2 compares the monthly averaged lightning flash rates accross different latitude bands in the NH, over land and ocean regions. The figure incorporates satellite observations (ISS-LIS and LIS/OTD) and results from two simulation experiments (LNOx-CTH and LNOx-ICEFLUX). Flash rates from the LNOx-CTH experiment show a clear seasonal cycle, with peaks occurring in May–August over NH and lands. During the winter months (November–January), flash rates drop significantly, being 5–7 times lower than the summer (May–August) peak values. The seasonal variation observed in the LNOx-ICEFLUX experiment and satellite observations closely align with this trend. Over land, both observations and simulations indicate high flash rates, particularly in the tropics, followed by midlatitudes for all the months. Peak lightning activity over lands occurs during late spring and early summer (May–August), corresponding to enhanced convective activity (Holle et al., 2016; Ghosh et al., 2023). In contrast, flash rates over oceans are consistently lower over all latitude bands. The tropical ocean shows an uniform flash rates throughout the year, without any prominent seasonality. The delay in the seasonal peak of flash rates over NH with CTH, and particularly with ICEFLUX, as noted by Finney et al. (2014), is not seen in our simulations. Therefore, using near-real-time, high spatially and temporally resolved meteorological data from ECMWF-IFS with continuous updates and improved configuration for advection in CHIMERE, we achieve an improved seasonal distribution that match well the satellite measurements.

The modelled monthly averaged flash rates from the simulations exhibit a strong positive correlation with satellite observations (ISS-LIS and OTD/LIS), with correlation coefficients ranging from 0.85 to 0.97 (Table 4). This agreement is consistent across all latitude bands over the NH, land regions, and the polar ocean region from both the experiments (LNOx-CTH and LNOx-ICEFLUX). These findings indicate that the simulations successfully capture the seasonal variability of flash rates."

4. L125: Typically, the "land-based" parameterization is used for grid boxes near land as well, i.e. continental convection extends a few grid boxes downwind of land.

**Reply:** Yes, but in our simulations for each grid-cell, the relative fraction of sea ($x_{sea}$) is determined using the land-sea mask from the land use database and fraction of land is estimated with ($1 - x_{sea}$). This method accounts for the influence of both land and sea in a grid-cell, on parameterized processes such as convection and lightning, providing more realistic simulations in coastal and near-land regions.

**Changes in the manuscript:** No changes are made.

5. L132: Could you give values of $H_f$ for 0, 30, and 60 N.

**Reply:** The cold cloud depth ($H_f$, from 0°C to cloud top) values at 0°, 30°N and 60°N latitudes are 7.5, 7.46 and 7.15 km, respectively. It is to be noted that $H_f$ (in km) in this study, is calculated with the temperature profile in the model, estimating the freezing temperature height or the freezing level. The modelled $H_f$ from our study varies from 6.9 to 7.76 km. The freezing level (0°C temperature), estimated in model, varies from 1 to 4.9 km, being the highest at tropics and decreasing with higher latitudes. The flashes from freezing level to the cloud top height is considered as 'in cloud (IC)' flashes and from freezing level to ground as 'cloud-to-ground (CG)' flashes. We have now added the details of estimated $H_f$ in the manuscript.

**Changes in the manuscript:** The details are now added in our manuscript. Please see the lines 151–158, "In this study $H_f$ (in km) is calculated with the temperature profile in the model, estimating the freezing temperature height or the freezing level. The modelled $H_f$ from our study varies from 6.9 to 7.76 km. The freezing level (0°C temperature), estimated in model, varies from 1 to 4.9 km, being the highest at tropics and decreasing with higher latitudes. The flashes from freezing level to the cloud top height is considered as 'in cloud (IC)' flashes and from freezing level to ground as CG flashes."

6. L135: What is the ratio of CG to IC flashes that you obtain over the NH with this parameterization?

**Reply:** The mean ratio of IC to CG flash rates ($\beta$ = IC/CG) is estimated as 3.09 in our model, varying within a range of 1.6–4.7. $\beta$ is estimated with the empirical equation developed by Price and Rind (1993) based on cold cloud depth ($H_f$). Here $H_f$ in this study, is estimated with the temperature profile in the model. The value of $\beta$ estimated in our study is comparable to that obtaibed in recent modelling studies (Luhar et al., 2021; Gordillo-Vázquez et al., 2019). Wu et al. (2023) estimates the values of $\beta$ as 2.94–3.70 with a lightning nitrogen oxides (LNOx) emissions model using satellite-observed lightning optical energy. Further experiments conducted with satellite- and ground-based observations over different parts of the world also produce a $\beta$ value in the range of 2.64–2.94 ($\pm$1.1–1.3) over US (Boccippio et al., 2001), 3–4 over India and China (Ghosh et al., 2023; Ren et al., 2024). $\beta$ obtained in our study again shows consistency with the above mentioned results.

**Changes in the manuscript:** The details are now added in our manuscript. Please see the lines 159–170, "The mean ratio of IC to CG flash rates ($\beta$ = IC/CG) is estimated as 3.09 from model estimates. $\beta$, in this study, is estimated with the empirical equation (Equation 1) by Price and Rind (1993), which is frequently used in several modelling studies (Luhar et al., 2021; Gordillo-Vázquez et al., 2019). The empirical relationship between $\beta$ and the cold cloud depth ($H_f$) was developed by Price and Rind (1993) based on data collected for 139 individual thunderstorms over the western United States (US) during summer. Several studies support the fact that Price and Rind (1993) parameterization successfully estimate the distribution of CG and

IC flashes in global as well as in mesoscale models (Pickering et al., 1998; Fehr et al., 2004). Theoretically, $\beta$ varies between 1 to 50 for $H_f$ varying between 5.5 to 14 km to prevent unrealistic values. The value of $\beta$ estimated in our study is comparable to that obtaibed in recent modelling studies (Luhar et al., 2021; Gordillo-Vázquez et al., 2019). Wu et al. (2023) estimates the values of $\beta$ as 2.94–3.70 with a lightning nitrogen oxides (LNOx) emissions model using satellite-observed lightning optical energy. Further experiments conducted with satellite- and ground-based observations over different parts of the world also produce a $\beta$ value in the range of 2.64–2.94 ($\pm 1.1$–1.3) over US (Boccippio et al., 2001), 3–4 over India and China (Ghosh et al., 2023; Ren et al., 2024). $\beta$ obtained in our study again shows consistency with the above mentioned results."

$$\beta = 0.021 H_f^4 - 0.648 H_f^3 + 7.49 H_f^2 - 36.54 H_f + 63.09 \tag{1}$$

7. L140: Include a plot showing the percent of CG and IC LNOx emissions input into each 1-km layer for land and water over the tropics and midlatitudes.

**Reply:** We thank the reviewer for suggesting the plot. The total LNOx emissions for each model level are available in our model output. The LNOx emissions as percentage mass per km are plotted for tropical and mid-latitudinal land and ocean (please see the Figure 3b–3e in manuscript).

**Changes in the manuscript:** We have added a section describing the Figures 3b–3e and providing the details. Please see the lines 379–396, "The Figures 3b–3e represent the vertical distribution of LNOx emissions as percentage of LNOx mass per km. The distribution shows the maximum of LNOx mass lies between the altitude range of 4–7 km at all regimes from both the simulations, showing the typical 'backward C-shape' (Ott et al., 2010). 60–65% of LNOx mass is injected at this altitude range. Here it is to be mentioned that annually 1.85 and 1.9 Tg N LNOx are being generated over tropical land as obtained from LNOx-CTH and LNOx-ICEFLUX simulations, respectively, being almost 63%–66% of total annual LNOx over NH. The amounts are 0.55 (0.76), 0.15 (0.09) and 0.07 (0.03) Tg N for mid-latitudinal land, tropical ocean and mid-latitudinal ocean respectively, from LNOx-CTH (LNOx-ICEFLUX) simulation. Therefore, the mid-tropospheric region (4–8 km) contributes the maximum of the LNOx mass, specially over the tropical land region. The vertical profiles available from previous studies reveal a similar shape of all the profiles but contributing maximum at upper tropospheric region (within 2–4 km of the tropopause) rather than mid-troposphere. However, the study by Pickering et al. (1998) represents a high emission near surface due to strong downdraft, where the distribution is low and almost uniform up to 5–6 km as observed from the studies by Ott et al. (2010); Luhar et al. (2021). Nevertheless, the profile over mid-latitude lands from our study matches well with that from the study by Ott et al. (2010), with a maximum at 5 km. In our study, 13%–19% of total LNOx mass is estimated from surface to up to 2 km over tropical and mid-latitude lands and tropical oceans. LNOx production is suggested to be proportional to atmospheric pressure by Goldenbaum and Dickerson (1993); Pickering et al. (1998). Notably, a simple vertical structure of the emissions is adopted in this study, considering the emissions to be evenly distributed over an altitude range. The vertical distribution of LNOx mass can be improved by replacing the simple distribution currently used, with the more detailed scheme developed by Pickering et al. (1998)."

8. L187: By what metric is the flash rate in agreement with the ISS-LIS data set? How was the ISS-LIS May-August "climatology" obtained? What years were used? Was there a correction for viewing time? Was there a correction for detection efficiency? How many total flashes are there during May-August (and the year) for ISS-LIS and the model. How do the flash totals compare to the OTD/LIS climatology for this period.

**Reply:** We have evaluated the simulated lightning flash rates against that obtained from ISS-LIS (domain: $\pm 55°$ latitudes) satellite measurements for the year 2018. The specific year is now mentioned in the manuscript (line no. 218 in Section 2.5). Daily data of lightning flash is obtained with high spatial and temporal resolution. We have estimated the monthly mean of flash rates from the data to compare with the simulated monthly averaged flash rates.

There was a correction in viewing time (Blakeslee et al., 2020). The ISS-LIS operates at a frame rate of 500 frames per second, corresponding to a native timing precision of 2 milliseconds. The actual on-orbit timing accuracy of the ISS-LIS was validated through comparisons with multiple ground-based and spaceborne reference datasets. Initially, timing data from the ISS-LIS showed offsets of up to ±1 second compared to reference data, with a cyclic drift pattern recurring approximately every 9 days. Through detailed analysis, this drift was characterized, and timing correction variables, along with a constant offset, were applied to improve timing accuracy. After corrections, done by comparing with Geostationary Operational Environmental Satellite (GOES) 16 and 17 Geostationary Lightning Mappers (GLM-16/17) and ground-based observations, the timing accuracy of ISS-LIS is better than its native precision of 2 ms.

Comparisons with the other sensors suggest a flash detection efficiency of ISS-LIS, is around 60% with diurnal variability of 51%–75% (Blakeslee et al., 2020).

As mentioned previously, we have now re-run the simulation with cloud top height (CTH) scheme applying modifications in the equations for estimating flash rates over lands and oceans, to reconcile the estimated flash rates with the satellite observations. We also have carried out simulations with an ice flux based lightning parameterization (ICEFLUX). The annual flash frequencies over the NH is estimated as 20.7 and 21.6 flashes s$^{-1}$, respectively from the simulations with CTH and ICEFLUX schemes, are in alignment to the satellite observations. The flash frequencies are 23.6 and 26.4 flashes s$^{-1}$, respectively from ISS-LIS and LIS/OTD climatology data (May, 1995 to December, 2014) over NH. The flash frequency estimated from LNOx-ICEFLUX is divided by a factor 5 to reconcile with the satellite-observed frequency.

We also have evaluated monthly flash rates, instead of the flash rates for the period of May–August, from both the simulations and compared to the satellite observed flash rates. The modelled monthly averaged flash rates from the simulations exhibit a strong positive correlation with satellite observations (ISS-LIS and OTD/LIS), with correlation coefficients ranging from 0.85 to 0.97. This agreement is consistent across all latitude bands over the NH and land regions, as observed from both the experiments (LNOx-CTH and LNOx-ICEFLUX). These findings indicate that the simulations successfully capture the seasonal variability of flash rates. Please refer the reply of question no. 3, for more details.

**Changes in the manuscript:** Please see the description of ISS-LIS in lines 218–224, "Flash rate for the year 2018 from Lightning Imaging Sensor (LIS) on the International Space Station (ISS) platform, is used for evaluating flash rate estimated with the model (http://ghrc.nsstc.nasa.gov/; last access: 5 July, 2024). ISS-LIS optically detects lightning flashes that occur within its field-of-view during both day and night with storm-scale (4 km × 4 km) horizontal resolution (Blakeslee et al., 2020) and 2 ms of temporal resolution. After time corrections comparing with Geostationary Operational Environmental Satellite (GOES) 16 and 17 Geostationary Lightning Mappers (GLM-16/17) and ground-based observations, the timing accuracy of ISS-LIS is better than its native precision of 2 ms. The flash detection efficiency of ISS-LIS is around 60% with diurnal variability of 51%–75% (Blakeslee et al., 2020)."

Lines 339–344 in manuscript for comparison of simulated flash rates to the satellite observations, "The annual flash frequencies over the NH is estimated as 20.7 and 21.6 flashes s$^{-1}$, respectively from the LNOx-CTH and LNOx-ICEFLUX experiments (Table 5). These values are consistent with the satellite observations (23.6 and 26.4 flashes s$^{-1}$, respectively from ISS-LIS and OTD/LIS observation over NH) as well as to that obtained in recent model-based studies over NH (Luhar et al., 2021). While no scaling factor is applied in the simulated flash rates from LNOx-CTH, the flash frequency from LNOx-ICEFLUX is divided by a factor 5 to reconcile with the satellite-observed frequency. Accordingly, the evaluation of the results from LNOx-ICEFLUX has been conducted using flash rates adjusted by this factor."

Also see the lines 328–331, "The modelled monthly averaged flash rates from the simulations exhibit a strong positive correlation with satellite observations (ISS-LIS and OTD/LIS), with correlation coefficients ranging from 0.85 to 0.97 (Table 4). This agreement is consistent across all latitude bands over the NH, land regions, and the polar ocean region in both the experiments (LNOx-CTH and LNOx-ICEFLUX). These findings indicate that the simulations successfully capture the seasonal variability of flash rates."

9. L189: Yes, putting too many flashes in the tropics is a fairly well known bias of cloud-height based schemes. Do you have any suggestions as to how this bias could be reduced while still retaining the cloud-height scheme?

**Reply:** As mentioned previously, we have now re-run the simulation with cloud top height (CTH) scheme applying modifications in the equations for estimating flash rates over lands and oceans, to reconcile the estimated flash rates with the satellite observations. We also have carried out simulations with an ice flux based lightning parameterization (ICEFLUX). Now the annual flash rates in tropical land and NH regions from both experiments (LNOx-CTH and LNOx-ICEFLUX), as well as in mid-latitudes from LNOx-ICEFLUX, show good agreement with satellite observations.

**Changes in the manuscript:** We have added a detailed comparison of simulated annual flash rates with satellite observations. Please see the lines 293–312, "Spatially mean annual flash rates over tropical lands from both the experiments are close enough to the satellite measurements at tropics over NH and lands, while at mid-latitudes, only a good resemblance is observed for that from LNOx-ICEFLUX. On the other hand, the simulation with CTH scheme estimates the flash rates over tropical ocean as almost twice of that estimated using the ICEFLUX scheme, unlike that showed in previous studies (Finney et al., 2016). We compare the spatially varying simulated annual flash rates with satellite observations (ISS-LIS and OTD/LIS) and the corresponding statistical metrics are presented in Table 3. Correlation coefficients for spatially varying flash rates show comparatively stronger correlations over the tropics and mid-latitudes of the NH and land regions, between that from LNOx-CTH and satellite data, compared to LNOx-ICEFLUX, being consistent with the findings by Clark et al. (2017). The flash rates from LNOx-CTH exhibit significantly higher correlations, particularly when evaluated against OTD/LIS data. Analysis of RMSE and NME also indicates lower errors for LNOx-CTH over these regions in comparison to that observed for LNOx-ICEFLUX, indicating the spatial variations of flashes from the first experiment align well with the satellite data. The worst performance is observed in the polar lands compared to the other two latitude bands, and also over oceanic region from both the experiments, characterized by weak correlations and higher errors. Hence, both schemes struggle to accurately simulate flash rates over oceans and high latitude lands. However, since lightning activity is minimal in these regions, the impact of this limitation is relatively minor. These results also highlight the ongoing challenges of accurately representing convection and capturing lightning flashes over the oceans. In summary, the statistical analysis points out the effectiveness of the LNOx-CTH scheme in reproducing the spatially varying lightning flash rates reasonably well, particularly at tropics over NH and lands. However, both parameterizations exhibit limitations in the polar regions and over oceans, indicating scopes for further improvement in the parameterizations. Despite its lower overall performance in capturing spatial variation of flashes, LNOx-ICEFLUX shows lower bias in magnitude over mid-latitudinal land regions, suggesting the potential for refinement."

*You need to add comparisons with OMI/TROPOMI and SHADOZ.*

10. L214: The impact of lightning-NOx on atmospheric composition is larger in the mid- and upper-troposphere than it is at the surface, yet you begin with a surface comparison. I would suggest saving that comparison for last.

**Reply:** We agree with the reviewer. We now first evaluated the impact of lightning on the vertical distributions of tropospheric $O_3$ and $NO_2$. Subsequently, we assessed its impact on the surface concentrations of $O_3$ and $NO_2$.

**Changes in the manuscript:** Section 3.3 represents 'Vertical distribution of gases: effect of LNOx' and Section 3.4 represents 'Impact of lightning on surface-level $O_3$ and $NO_2$'.

11. You compare with surface concentrations and profiles. You should also compare model $NO_2$ with satellite-retrieved columns from OMI (January–December 2018) or TROPOMI (April–December 2018).

**Reply:** We thank the reviewer for the suggestions. We have now incorporated a comparison of modelled $NO_2$ with satellite-retrieved columns from Ozone Monotoring Instrument (OMI) for the year 2018 (January–December). A new Subsection 3.3.2 has been added.

**Changes in the manuscript:** Please see the lines 482–493 in manuscript, "The zonally averaged $NO_2$ column distribution (Figure 10) reveals elevated column densities over the tropics, especially between 20°–30°N, and the mid-latitudes, even in the absence of LNOx emissions. The zonal averages range from 0.35–1.75 $\times$ $10^{15}$ molecules $cm^{-2}$ in these regions, which is nearly double the values observed at higher latitudes (60°–90°N). The peak at 20°–30°N is, however not observed in satellite observations. The higher uncertainty in tropospheric $NO_2$ columns retrieved from OMI observations is attributed to the complexity of tropospheric processes and the sensitivity of retrievals towards pollution levels (Bucsela et al., 2013). The simulations also identify a secondary maximum in $NO_2$ column density at 35°–45°N, which aligns with satellite observations over the mid-latitudes. At higher latitudes, where the magnitudes are comparatively lower, the simulated $NO_2$ column density matches well with satellite-based observations. Overall, the zonally averaged $NO_2$ column densities from the simulations closely replicate satellite observations, except for a pronounced peak at 20°–30°N from simulated $NO_2$. The tropospheric burden of $NO_2$ is 146 Gg from LNOx-CTH, being comparable to that from noLNOx and 11% lower than that estimated from OMI (Table 8). The burden estimated from LNOx-ICEFLUX is 18% and 5% higher than the LNOx-CTH and OMI, respectively."

12. The SHADOZ ozonesondes are a wonderful source of profile information and should be compared to. Despite the acronym, there are a number of Northern Hemisphere sites.

**Reply:** Yes, we now have compared $O_3$ vertical profile with the same obtained from SHADOZ ozone-sondes at stations Kuala Lampur, Hanoi, Costa Rica and Hilo, situated over the NH tropics.

**Changes in the manuscript:** Please see the lines 439–445 in manuscript, "We also compare the vertical profiles of simulated $O_3$ with that from ozone-sonde measured data from Southern Hemisphere ADditional OZonesondes (SHADOZ) at stations Kuala Lampur, Hanoi, Costa Rica and Hilo, situated over the NH tropics. The plots are provided in supplementary material (Figure S2). Among the stations, at Kuala Lampur, the simulated $O_3$ profile from LNOx-ICEFLUX shows a good match with the observations from 2 km upwards, while the profile from LNOx-CTH aligns well with observations at Costa Rica. However the modelled $O_3$ profiles show under- and overestimation at most of the altitudes, respectively at Hanoi and Hilo, but overall replicate the observed altitudinal distribution quite well. The comparisons once again represents the effect of LNOx on $O_3$, specifically at the free troposphere."

It should be more revealing to calculate biases as a function of season as opposed to annually.

**Reply:** Yes, we now have evaluated monthly $O_3$ for the altitude bands of 750–500 hPa and 500–200 hPa, with respect to the WOUDC ozone-sonde data.

**Changes in the manuscript:** Please see the lines 446–454 in manuscript, "The monthly comparison of simulated $O_3$ from the LNOx-CTH and LNOx-ICEFLUX experiments with WOUDC ozone-sonde measurements, is presented in Figure S2 in the supplementary material, across two altitude (750–500 hPa and 500–200 hPa) and three latitude bands. The monthly variation reveals the highest peaks during March and October in the tropics, with the lowest levels observed during June–July for both altitude bands. A similar variation is noted at mid-latitudes and in the polar regions for the 750–500 hPa altitude band, aligning well with the observed monthly trends. The simulated $O_3$ from both experiments closely matches the observed values at the tropics and mid-latitudes for both altitude bands, exhibiting a low bias of $\pm3$–10 ppbv. However, an overestimation of simulated $O_3$ is apparent during January–April, particularly over the tropics. While there is a good agreement between simulated and observed $O_3$ over the polar region for the 750–500 hPa altitude band, a significant underestimation of simulated $O_3$ is evident in the 500–200 hPa altitude band as discussed previously."

*You need to improve the model simulation. This will add value to the comparison with observations.*

13. L315-320: $CH_4$ lifetime of 4.89 years is considerably less than range of 7–14. Obviously, your large LNOx source is contributing to this low bias in lifetime.

**Reply:** No, the underestimated $CH_4$ lifetime in model is not due to high LNOx emissions in this study. The underestimated $CH_4$ lifetime is even estimated from the simulation without including LNOx. It is due to the higher magnitude of OH concentrations, which promote the oxidation of $CH_4$, reducing the burden and increasing the chemical loss of $CH_4$, even in the absence of LNOx. Therefore, the underestimated $CH_4$ lifetime in the present study, likely stems from other factors including issues related to deficiencies in the chemistry or photolysis schemes. Addressing and resolving this issue will require further investigation in future studies.

**Changes in the manuscript:** Please see the lines 556–560 in manuscript "The underestimation in $CH_4$ lifetime in our study is due to the overestimation of OH concentrations, which promote the oxidation of $CH_4$, reducing the burden and increasing the chemical loss of $CH_4$, even in the absence of LNOx. Therefore, the underestimated $CH_4$ lifetime in the present study is not attributed to LNOx but likely stems from other factors including issues related to deficiencies in the chemistry or photolysis schemes. Addressing and resolving this issue will require further investigation in future studies."

*You need to be careful with generalized statements.*

14. There are numerous reasons why a model may have a high-or-low bias in ozone at the surface. The magnitude of the LNOx source plays only a small role here.

**Reply:** Yes, there are numerous reasons to show high or low bias in simulated $O_3$ at surface. Uncertainty in input emissions to the model is one of such reasons and LNOx is one of such uncertain emission. We have now removed the line to avoid confusions.

**Changes in the manuscript:** We have removed the line.

15. Stratosphere-troposphere exchange is poorly resolved in this model due to its low model top. This may make comparison with mid- and high-latitude ozone profiles problematic.

**Reply:** We have now observed an improved vertical distribution of $O_3$ due to use of van Leer (1977) scheme for vertical transport instead of the anti-diffusive scheme of Després and Lagoutière (1999). The bias in simulated $O_3$ at the free troposphere is now much reduced than we estimated previously using Després and Lagoutière (1999) as the vertical transportation scheme as presented in the previous version of the manuscript (https://doi.org/10.5194/egusphere-2024-3087). But still high underestimation in the simulated $O_3$ mixing ratio exists in the altitude band 500–200 hPa, i.e., the upper troposphere and lower stratosphere, over mid-latitudes and polar regions. The bias is observed for both the simulations with LNOx, however lower underestimation is observed for LNOx-ICEFLUX. The underestimation represents that improvement in the modelled stratosphere-troposphere exchange still requires a lot of refinement and further investigation.

**Changes in the manuscript:** Please see the line 102, "The horizontal and vertical transports are solved with the van Leer (1977) scheme."

Also refer to the lines 435–438, "The high underestimation in the simulated $O_3$ mixing ratio in the altitude band 500–200 hPa, i.e., the upper troposphere and lower stratosphere, over mid-latitudes and polar regions still exists, even after the inclusion of LNOx, however lower underestimation is observed for LNOx-ICEFLUX. The underestimation represents that improvement in the modelled stratosphere-troposphere exchange still requires a lot of refinement and further investigation."

**Minor Comments:**

16. L24: The range you cite here (8 to 4000 moles NO / flash) is for individual flashes. Estimates of the mean production per flash are usually between 125 and 500 moles per flash.

**Reply:** We thank the reviewer for the correction. We have now modified the values.

**Changes in the manuscript:** Please see the lines 26–28, "The mean estimated rate of NOx emissions caused by lightning is highly uncertain, with recent studies indicating variations ranging between 33–660 mol NO per flash (Luhar et al., 2021; Bucsela et al., 2019; Murray, 2016; Schumann and Huntrieser, 2007), although Schumann and Huntrieser (2007) suggest a mean value of 250 mol NO per flash."

17. L73: Is 998 hPa the mean surface pressure?

**Reply:** No, 998 hPa is not the mean surface pressure, the mean sea level pressure is around 1013 hPa. In this study, 998 hPa represents the pressure at the top of the lowest $\sigma$-pressure level, representing near-surface conditions. We have now removed the 998 hPa to avoid confusion.

**Changes in the manuscript:** Please see the lines 86–87, "Simulations are done in twenty vertical levels in $\sigma$-pressure coordinates ranging from surface to 200 hPa for a period of one year (January–December, 2018) with a spin-up time of 15 days."

18. L74: A 200 hPa top layer is when evaluating $O_3$ profiles. What is the pressure of the model top? Clearly, this model was designed for tropospheric chemistry.

**Reply:** The pressure at the CHIMERE model top is 200 hPa. Yes, the model is primarily designed for tropospheric chemistry. Boundary and initial conditions (derived from Copernicus Atmosphere Monitoring Service (CAMS) reanalysis dataset of atmospheric compositions produced by ECMWF for this study) also takes the stratospheric chemistry into account.

**Changes in the manuscript:** No changes are made.

19. L160 and L201: Is this a climatological flash data set from ISS-LIS? If so, what years does it cover?

**Reply:** The data we have used in our study is not a climatological data. We have used a single year data for the year 2018. The specific year is now mentioned in the manuscript.

**Changes in the manuscript:** Please see the lines 218–219, "Flash rate for the year 2018 from Lightning Imaging Sensor (LIS) on the International Space Station (ISS) platform, is used for evaluating flash rate estimated with the model."

20. L192: Are you saying that the modelled tropopause height is too high in the tropics? If yes, please show.

**Reply:** Yes, the average model tropopause height is about 15.6 km in tropics, which is higher than the model cloud top height averaged over the tropics (11.6). Table S1 is now added in supplementary material presenting the topopause and cloud top height. We also have discussed it in manuscript.

**Changes in the manuscript:** Please see the lines 289–292, "In CTH scheme, the estimated flash numbers are dependent on the cloud top height, which is limited by the tropopause height. The tropopause height is highest at equator and decreases away from the equator (refer to Table S1 in supplementary material presenting the tropopause and cloud top height). This restricts the cloud development to lower altitudes, resulting in a significant reduction in flash rates at higher latitudes (Finney et al., 2014)."

21. L200: Why are you comparing May-August flash rates with the annual mean?

**Reply:** This is explained in the reply of question no. 3 of major comments. We have removed the comparison of flash rates during May–August with the annual mean.

Currently, with improved configurations for advection in the model, we have conducted the simulations (for the year 2018) to estimate lightning flashes with CTH and ICEFLUX schemes. We have evaluated monthly flash rates from both the simulations and compared to the satellite observed flash rates from ISS-LIS (for the year 2018) and LIS/OTD (monthly climatology data for the priod of May, 1995 to December, 2014).

**Changes in the manuscript:** Please see the changes provided in question no. 3 of major comments.

22. L207: You need to be clear here. 8.8 Tg N yr-1 for the NH is consistent with 15 Tg N yr-1 for the globe, a value that is well outside of the 2-8 range but below 25.

**Reply:** This is already explained in question no. 1 of major comments.

**Changes in the manuscript:** Please see the changes provided in question no. 1 of major comments.

23. L214: Why do you think emissions are underestimated?

**Reply:** Traditionally, emission inventory database is formed from the 'bottom-up' approach based on activity data and emission factors. Unavailability of precise activity data and lack of information on field-measured emission factors with fuel type and combustion conditions impose uncertainty in the emissions which is applied as input to the models.

**Changes in the manuscript:** We have removed the line.

24. L265: Do any of these studies give the NH burden? Or do they just list the total burden?

**Reply:** The references mentioned here provide the global burden. Now we have removed these references and added a comparison with the burden estimated with data obtained Aura Ozone Monitoring Instrument/Microwave Limb Sounder (OMI/MLS) for the 0°–60°N domain. The tropospheric $O_3$ burdens are 176 (150) Tg and 182 (155) Tg, respectively estimated from the simulation LNOx-CTH and LNOx-ICEFLUX over the NH and over the domain of 0°–60°N (presented inside parenthesis). These burdens represent a 7%–11% increase relative to the noLNOx simulation (Table 8). Notably, the estimated $O_3$ burden in this study aligns closely with observations from Aura Ozone Monitoring Instrument/Microwave Limb Sounder (OMI/MLS) for the 0°–60°N domain (159 Tg).

**Changes in the manuscript:** Please see the lines 462–465, "The tropospheric $O_3$ burdens, estimated from the simulation LNOx-CTH and LNOx-ICEFLUX, are respectively 176 (150) Tg and 182 (155) Tg, over the NH and over the domain of 0°–60°N (presented inside parenthesis). These burdens represent a 7%–11% increase relative to the noLNOx simulation (Table 8). Notably, the estimated $O_3$ burden in this study aligns closely with observations from OMI/MLS (159 Tg) for the domain of 0°–60°N."

25. L269: Are you comparing to climatological ozone sondes or ozone sondes from 2018? You should consider comparing to individual SHADOZ sondes as several of the sites are located in NH.

**Reply:** We have compared the $O_3$ vertical profile with the ozone-sonde data obtained from the World Ozone and Ultraviolet Radiation Data Centre (WOUDC), for a single year of 2018. The year is now added in manuscript. We also have compared $O_3$ vertical profile with SHADOZ ozone-sondes and presented.

**Changes in the manuscript:** Please see the lines 229–235, "For evaluating the vertical profile of $O_3$, altitudinal data measured by ozone-sonde, launched on small balloons, are downloaded from the World Ozone and Ultraviolet Radiation Data Centre (WOUDC, https://woudc.org/data, last access: 5 July, 2024) for the year 2018. Ozone-sonde data from 122, 977 and 121 stations are collected, respectively, over the tropical (0°–30°N), mid-latitudes (30° N–60° N) and polar region (60° N–90° N). We also have used vertical $O_3$ profile data from Southern Hemisphere ADditional OZonesondes (SHADOZ) ozonesonde measurements (https://tropo.gsfc.nasa.gov/shadoz, last access: 21 November, 2024) at four tropical stations (Kuala Lampur: 3.14°N, 101.69°E; Hanoi: 21.02°N, 105.80°E; Costa Rica: 9.62°N, −84.25°E and Hilo: 19.72°N, −155.08°E) for the year 2018."

26. L279: Stratosphere-troposphere exchange is one cause for the very low biases you are seeing in the extratropics and at high latitudes.

**Reply:** This is already explained in question no. 15 in major comments.

**Changes in the manuscript:** Changes are provided in question no. 15 in major comments.

**Grammatical Comments:**

27. L19: such as lightning — such as lightning and soil-NOx emissions

**Reply:** This is done.

**Changes in the manuscript:** Please see the lines 20–21, "NOx emissions arise from both anthropogenic sources, e.g., fossil fuel combustion, biomass burning, and natural processes, such as lightning and soil-NOx emissions (Verma et al., 2021; Butler et al., 2020)."

28. L24: 4000 mol NO per flash — 4000 mol NO per individual flash

**Reply:** This is modified.

**Changes in the manuscript:** Please see the lines 26–28, "The mean estimated rate of NOx emissions caused by lightning is highly uncertain, with recent studies indicating variations ranging between 33–660 mol NO per flash (Luhar et al., 2021; Bucsela et al., 2019; Murray, 2016; Schumann and Huntrieser, 2007)"

29. L25: suggest a value of 250 — suggest a mean value of 250

**Reply:** This is done.

**Changes in the manuscript:** Please see the lines 28, "although Schumann and Huntrieser (2007) suggest a mean value of 250 mol NO per flash."

30. L42: most frequently used CTH scheme — the frequently used CTH scheme

**Reply:** We have modified the lines.

**Changes in the manuscript:** Please see the lines 45–46, "In this study, we expand on the previous work by implementing the scheme based on ice flux (ICEFLUX) in CHIMERE and comparing it with the CTH scheme."

31. L75: for chemical mechanisms — for the chemical mechanism

**Reply:** This is done.

**Changes in the manuscript:** Please see the lines 87–88, "The MELCHIOR2 scheme is used for the chemical mechanism."

32. L91: deep convection fluxes — deep convective fluxes

**Reply:** This is done.

**Changes in the manuscript:** Please see the lines 102–104, "Boundary layer height and vertical diffusion are calculated by the parameterization proposed by Troen and Mahrt (1986) and deep convective fluxes are estimated using the Tiedtke (1989) scheme."

33. L160: ... but ISS-LIS only sees a small subset of total flashes ...

**Reply:** The flash detection efficiency of ISS-LIS is around 60% with diurnal variability of 51%–75% (Blakeslee et al., 2020). We have added the details about ISS-LIS.

**Changes in the manuscript:** Please see the lines 223–224, "The flash detection efficiency of ISS-LIS is around 60% with diurnal variability of 51%–75% (Blakeslee et al., 2020). "

34. L197: inconsistancies — inconsistencies.

**Reply:** This is corrected.

**Changes in the manuscript:** Please see the lines 278–279, "Previous studies have reported inconsistencies in the equation for oceanic flashes developed by Price and Rind (1992) (Michalon et al., 1999; Boccippio, 2002; Luhar et al., 2021)."

35. L223: represents a spatial variation — varies spatially

**Reply:** This is done.

**Changes in the manuscript:** Please see the lines 500, "In our study, the $O_3$ mixing ratio at the surface varies spatially "

36. L225: rest part of NH — rest of the NH

**Reply:** This is done.

**Changes in the manuscript:** Please see the lines 500–502, "In our study, the $O_3$ mixing ratio at the surface varies spatially with higher values ranging between 35–45 ppbv over the latitude band of 10°–50°N, specifically over the lands, being almost 1.5–2 times of that observed over the rest of the NH."

37. L233: 10-25 ppbv enhancement of annual mean surface ozone due to lightning –

**Reply:** The values are now lowered as estimated with new simulations.

**Changes in the manuscript:** Please see the lines 508–510, "A comparatively larger increase by 3–5 ppbv is observed over tropical parts of America and Africa and the Tibetan Plateau, but is particularly noteworthy (5–10 ppbv) over the central part of Africa"

38. Figure 3d: 500-200 hPa or 500-300 hPa. Caption is not consistent with figure.

**Reply:** This is corrected.

**Reviewer 2**

I share the concerns of the reviewer who posted a comment on Nov. 19th, so I will not repeat them here.

One additional major concern: The scientific content of the paper is very low; it is not clear to me what new insights were found as a result of this analysis relative to what is already in the scientific literature. Therefore, I recommend that the authors consider publishing their manuscript in Geoscientific Model Development (https://www.geoscientific-model-development.net/) instead of ACP. GMD is meant for description papers of significant model development.

**Reply:** We sincerely thank the reviewer for their valuable suggestions. We have now re-run the simulations with parameterization based on cloud top height (CTH), applying modifications, to better align the modelled flash rates over lands and oceans to the satellite observations. Additionally, flash rates are computed using an updated ice flux based lightning scheme (ICE-FLUX) for a comparative study. We perform a detailed evaluation of modelled lightning flashes, simulated ozone and trace gases, integrating in situ measurements and satellite observations, to critically assess the reliability and applicability of these parameterizations.

Therefore, with the new improved simulations and detailed evaluation of the results, we believe that the scientific values of the manuscript has been increased to meet the standard of the journal 'Atmospheric Chemistry and Physics'.

**References**

[revised manuscript text omitted]

---

## Author Response (AR2)

**Review replies for the manuscript entitled "Representing improved tropospheric ozone distribution over the Northern Hemisphere by including lightning NOx emissions in CHIMERE" (egusphere-2024-3087) by Sanhita Ghosh et al.**

We sincerely thank the editor and the reviewer for the constructive feedback and insightful comments on our manuscript. Below, we address each comment in detail and outline the corresponding revisions made to improve the manuscript.

**Reviewer 1**

**Summary**

The authors implement a cloud-height based flash rate scheme and an ice-flux based scheme into the regional model CHIMERE and then evaluate the impact of lightning-NOx production on ozone profiles, the tropospheric OH burden, the methane lifetime, and ambient ozone and $NO_2$ concentrations. This version of the paper is much improved but remains unpublishable until the authors explain how the addition of LNOx leads to a decrease in $NO_2$ columns over much of the domain.

**Reply:** We thank the reviewer for the valuable feedback. In response to the concern regarding the decrease in the $NO_2$ column due to inclusion of LNOx, we have provided a detailed explanation of how the inclusion of LNOx emissions in our model influences the $NO_2$ levels. We have also updated the manuscript with additional plots to better illustrate and clarify this effect. We also have thoroughly responded to each comment and highlighted the revisions made to enhance the manuscript.

**Major Comments:**

1. L281: Be specific as to what is improved in this simulation over water versus the original PR92. My hunch is that you obtained a better ocean-simulation because the ratio of a/b you use is smaller than that used in PR92, thus after scaling a higher fraction of flashes are over water lessening the low-bias others have observed with PR92 over water. – Do you agree?

**Reply:** We have now specified the improvement made in our simulation with the scheme based on cloud top height (CTH) in respect to the original scheme by Price and Rind (1992), due to which we have obtained an improved flash rate over the ocean. Yes, our study demonstrates an improved flash rates over tropical oceans, using the CTH scheme with a correction factor of 0.5 applied to the constant 'b' in Equation 1 (please see Section 2.2.1), for the oceanic grids. A simulation with the original scheme by Price and Rind (1992) showed an overestimated flash rates over the tropical ocean by a factor ≈2, which is lowered to 1.3–1.4, after application of the correction factor in the present study.

**Changes in the manuscript:** Please see the lines 293–296, "However, our study demonstrates an improved flash rate distribution over tropical oceans, using the CTH scheme with a correction factor of 0.5 applied to constant 'b' in Equation 1, for the oceanic grids. A simulation with the original scheme by Price and Rind (1992) showed an overestimated flash rates over the tropical ocean by a factor ≈2, which is lowered to 1.3–1.4, after application of the correction factor in the present study."

2. L289-296: I don't follow your rationale here. Are you saying that more realistic midlatitude flash rates could be produced if cloud heights were allowed to extend above the tropopause? Shouldn't cloud heights be limited by the tropopause? Yes,

overshooting storms exist and do transport trace gases into the stratosphere but aren't anvils typically limited by the tropopause. Perhaps the heights of tropical storms are too high.

**Reply:** Here we intend to highlight the limitation of using cloud top height as the sole parameter for estimating lightning flash rates. While CTH scheme provides an useful approximation, since deeper convection generally correlates with higher lightning activity, it likely doesn't capture the full complexity of the processes driving lightning generation. Overshooting tops are a clear example of where relying solely on cloud top height can fall short, as these strong convective bursts can extend above the tropopause and influence lightning rates, even if the average cloud top height remains lower. Moreover, factors, such as updraft strength, cloud depth, ice water content and mixed-phase regions play critical roles in charge separation and lightning production. By strictly capping cloud heights at the tropopause in the CTH scheme, the model may indeed underestimate flash rates in the mid-latitudes. This highlights the need to consider a multi-parameter approach for estimating flash rates, incorporating updraft dynamics, cloud microphysics and ice-phase processes alongside cloud top height.

We have modified the lines for more clarity.

**Changes in the manuscript:** Please see the modified lines 304–310, "While CTH scheme provides an useful approximation, since deeper convection generally correlates with higher lightning activity, it likely doesn't capture the full complexity of the processes driving lightning generation. Factors, such as updraft strength, cloud depth, ice water content and mixed-phase regions play critical roles in charge separation and lightning production. By strictly capping cloud heights at the tropopause in the CTH scheme, the model may indeed underestimate flash rates in the mid-latitudes. This highlights the need to consider a multi-parameter approach for estimating flash rates, incorporating updraft dynamics, cloud microphysics and ice-phase processes alongside cloud top height."

3. Figure 5: Your scale does not include negatives or is there room for negatives in the point of the arrow? I'm surprised that $O_3$ from LNOx-ICEFLUX exceeds ozone from LNOX-CTH everywhere given the similarity in their total flash rates. Please double check your calculations and also verify that the legend matches what is plotted.

**Reply:** We have now modified the Figure 5 by changing the scale including negative values. The $O_3$ mixing ratio from LNOx-ICEFLUX is higher than that obtained from LNOx-CTH over most of the NH except few regions over the ocean. This is probably because of the higher LNOx emissions obtained in LNOx-ICEFLUX than the LNOx-CTH simulation. The $O_3$ mixing ratio from LNOx-ICEFLUX shows the maximum increase over the central Africa, followed by other tropical region for all the altitude bands. However, the increase in LNOx-ICEFLUX in respect to LNOx-CTH is smaller over most of the NH. Overall ther $O_3$ burden over NH from LNOx-ICEFLUX is 3% higher than that estimated from LNOx-CTH, while the increase in $O_3$ burden is 7%–11% with respect to the simulation without LNOx (noLNOx).

**Changes in the manuscript:** Please see the Figure 5 with modified scale.

4. L475-479: I don't understand how the inclusion of LNOx can lead to a decrease in column $NO_2$ over most of your non-tropical domain. This seems wrong. Please double check your calculations. How do the changes vary seasonally? Do you see increases during the spring and summer? Yes, you can see local decreases especially in the boundary layer over polluted regions but total columns must increase.

**Reply:** In our study, the decrease in $NO_2$ column density (in molecules $cm^{-2}$) due to inclusion of NOx from lightning, is primarily observed over the regions with high $NO_2$ pollution, such as southern and eastern Asia (India and eastern China), north-west Europe and eastern part of USA. The inclusion of LNOx in model increases large-scale ozone and OH concentrations, therefore reducing the lifetime of NOx through oxidation reactions with HOx including OH (Labrador et al., 2005; Schumann and Huntrieser, 2007). Figure S5 in supplementary material depicts the increase in $HNO_3$ column density over the above mentioned region, supporting the fact that $NO_2$ is oxidized and converted to the $HNO_3$, increasing the column density of $HNO_3$. Hence, rapid conversion of $NO_2$ into other compounds, such as $HNO_3$ leads to its subsequent removal and a net

decrease in $NO_2$ column density over the regions with high anthropogenic pollution. A decrease in $NO_2$ column density over the above mentioned regions, is also observed during the late spring and summer season (May–August).

The Figure S6 in supplementary material, showing changes in annual mean $NO_2$ mixing ratio (in ppbv) from experiment LNOx-CTH with respect to noLNOx, demonstrates a decrease in $NO_2$ by 0.1–0.3 ppbv over the regions with higher anthropogenic $NO_2$ pollution as mentioned above, at the altitude band 998–900 hPa, i.e., mostly near surface, followed by the altitude band 900–750 hPa. A very small increase (0.05 ppbv) is observed over most of the NH at the higher altitude bands (750–500 and 500–200 hPa), due to inclusion of LNOx emissions. Overall the $NO_2$ column density decreases over the regions with high anthropogenic pollution.

**Changes in the manuscript:** Please see the lines 512–527, "A decrease in $NO_2$ column density (0.2–0.6 $\times$ $10^{15}$ molecules $cm^{-2}$) due to inclusion of LNOx emissions, is primarily observed over the above mentioned regions with high $NO_2$ pollution (Figure 9b). The inclusion of LNOx in model increases large-scale $O_3$ and OH concentrations, therefore reducing the lifetime of NOx through oxidation reactions with HOx including OH (Labrador et al., 2005; Schumann and Huntrieser, 2007). Figure S5 in supplementary material depicts the increase in $HNO_3$ column density over the above mentioned region, supporting the fact that $NO_2$ is oxidized and converted to the $HNO_3$, increasing the column density of $HNO_3$. Hence, rapid conversion of $NO_2$ into other compounds, such as $HNO_3$, leads to its subsequent removal and a net decrease in $NO_2$ column density over the regions with high anthropogenic pollution. The Figure S6 in supplementary material, showing changes in annual mean $NO_2$ mixing ratio (in ppbv) from experiment LNOx-CTH with respect to noLNOx, demonstrates a decrease in $NO_2$ by 0.1–0.3 ppbv over the regions with higher anthropogenic $NO_2$ pollution as mentioned above, at the altitude band 998–900 hPa, i.e., mostly near surface followed by the altitude band 900–750 hPa. A very small increase (0.05 ppbv) is observed over most part of NH at the higher altitude bands (750–500 and 500–200 hPa), due to inclusion of LNOx emissions. Overall the $NO_2$ column density decreases over the regions with high anthropogenic pollution."

**Minor Comments:**

5. L88: 20 vertical levels is quite coarse and a 200 hPa model top Is very low.

**Reply:** Currently the simulations are conducted with twenty vertical levels in $\sigma$-pressure coordinates ranging from surface to 200 hPa. Yes, the vertical resolution is coarse and model top is low. Although, boundary and initial conditions (derived from Copernicus Atmosphere Monitoring Service (CAMS) reanalysis dataset of atmospheric compositions produced by ECMWF for this study) takes the stratospheric chemistry into account.

6. L125-130: As equation (2) is not used, to avoid confusion, this section could be shorted with the equation removed

**Reply:** The Equation 2 has been used to estimate the scaling factor (C), which is further used in Equation 3 to estimate the total flash rate ($F_{CTH}$). The scaling factor is used to adapt the Equation 1 to various model resolutions.

Please see the Section 2.2.1 and Equations 1, 2 and 3.

$$F_l = a \times H_{top}^{4.9}$$
$$F_o = b \times H_{top}^{1.73}$$

(1)

Price and Rind (1994) formulated an equation to adapt the above equations to various model resolutions. Here, the product of longitude and latitude resolution, denoted as $\Delta x \times \Delta y$, is measured in degrees$^2$.

$$C = 0.97241 e^{0.048203 \times \Delta x \times \Delta y} \tag{2}$$

The total flash rate ($F_{CTH}$) is then calculated as follows:

$$F_{CTH} = \frac{C \times (x_{\text{sea}} \times F_o + (1 - x_{\text{sea}}) \times F_l)}{25} \tag{3}$$

7. L156-157: Why is this distinction made? Why is it important that flashes above the freezing level are considered IC and flashes below it CG?

**Reply:** Cloud to ground (CG) flashes are typically associated with strong charge separation within the cloud, often involving a negative charge at the base of the cloud and a positive charge on the ground or vice versa (Dwyer and Uman, 2014). This type of lightning requires a large electric potential difference between the cloud and the ground, and this process is generally more common below the freezing level. In cloud (IC) flashes occur within the cloud are often associated with the discharge of charge between different regions inside the cloud. Above the freezing level, the presence of ice particles contributes to the charge separation mechanism. The interaction between ice and liquid water are more prominent in the upper troposphere where the temperature is below freezing. The freezing level acts as a natural boundary between the upper and lower parts of the cloud. Above the freezing level, ice particles contribute to the development of IC lightning, while below it, the atmosphere is typically in a liquid state, with the warmer environment aiding in the development of CG lightning.

We have added these details in the manuscript.

**Changes in the manuscript:** Please see the line 163–166, "The freezing level acts as a natural boundary between the upper and lower parts of the cloud. Above the freezing level, ice particles contribute to the development of IC lightning, while below it, the atmosphere is typically in a liquid state, with the warmer environment aiding in the development of CG lightning (Dwyer and Uman, 2014)."

8. L170: "Beta" = IC/CG = "formula"

**Reply:** This is done.

**Changes in the manuscript:** Please see the line 178, "$\beta$ = IC/CG (Equation 7), obtained in our study again shows consistency with the above mentioned results."

9. L201-202: Justify why you also include $NO_2$ emissions from lightning.

**Reply:** Lightning is one of the largest natural sources of NOx, including both NO and $NO_2$ in the atmosphere. During a lightning strike, the extremely high temperature causes the dissociation of nitrogen ($N_2$) and oxygen ($O_2$) molecules in the air into atomic nitrogen (N) and oxygen (O). These free radicals then recombine to form NO, which can be converted to $NO_2$ in the presence of oxygen (Murray et al., 2012; Murray, 2016; Finney et al., 2014). Lightning generate $NO_2$ with $NO_2$/NOx ratio $\approx$0.1 to 0.5 (Schumann and Huntrieser, 2007). Therefore, it is important to include $NO_2$ emissions also.

**Changes in the manuscript:** Please see the line 210–212, "Lightning generate $NO_2$ with $NO_2$/NOx ratio varying from 0.1 to 0.5 (Schumann and Huntrieser, 2007) Therefore, it is important to include $NO_2$ emissions also."

10. L260-262 and L341-344: Do you also divide by 5 in the tables? Assuming yes, introduce the factor of 5 in section 2.2.2. Move discussion in 341-344 2.2.2 too. You do apply scaling factors to LNOx-CTH but before the model was run.

**Reply:** Yes, all the calculations and statistical analyses in tables are conducted for the flash rates estimated with ICEFLUX scheme, divided by the factor 5. We have now included this information in Section 2.2.2.

Yes, we have applied scaling factors to LNOx-CTH before conducting the simulation as described in Section 2.2.1, and no scaling factor is applied in the simulated flash rates from LNOx-CTH. This is mentioned in the lines 361–362 "While no scaling factor is applied in the simulated flash rates from LNOx-CTH, the flash frequency from LNOx-ICEFLUX is divided by a factor 5 to reconcile with the satellite-observed frequency.".

**Changes in the manuscript:** Please see the lines 155–156 added to Section 2.2.2, "The estimated flash frequency from LNOx-ICEFLUX has been scaled down by a factor of 5 to align with satellite-observed frequencies. Consequently, the evaluation of LNOx-ICEFLUX results has been carried out using these adjusted flash rates."

11. L274: where in the United States are you referring to?

**Reply:** We mean the central and south-eastern part of the United States. This is now added in the manuscript.

**Changes in the manuscript:** Please see the lines 286–288, "However, patches of high flash rate observed in satellite data over central Canada, central and south-eastern part of the United States, central European countries, and northern Russia are not reflected in the modelled flash rates from either experiment."

12. L288: "explaining the effectiveness of ICEFLUX scheme over CTH, in capturing flashes over the midlatitudes". Yes, the mean flash rates from the ICEFLUX scheme appear to be more realistic in the midlatitudes. However, the error statistics you show in Table 3 still favor the CTH scheme. Why is that? Is there a metric for which the ICEFLUX scheme does better in the midlatitudes?

**Reply:** As presented in Table 2 and discussed in the manuscript, the mean flash rate from LNOx-ICEFLUX over the mid-latitudes closely aligns with satellite observations and is nearly twice that estimated from LNOx-CTH. This highlights the effectiveness of the ICEFLUX scheme in capturing a more realistic magnitude of flash rates over the mid-latitudes.

However, Table 3 presents the statistical evaluation of the spatial variability of annual flash rates, comparing modelled flash rates with ISS-LIS and LIS/OTD satellite observations on a grid scale. This statistical scores represent the efficiency of the CTH scheme in reproducing the spatial variation of lightning flashes reasonably well than that observed for ICEFLUX scheme. While LNOx-ICEFLUX provides a reasonable estimate of flash rate magnitudes over both the tropics and mid-latitudes, it struggles to accurately capture the observed spatial pattern of lightning activity.

The ICEFLUX scheme offers significant advantages over CTH for estimating lightning flash rates as it directly accounts for the microphysical and thermodynamic processes that drive charge separation in convective storms. While CTH is often used as a proxy for convective intensity, it does not explicitly capture the ice-phase interactions crucial for electrification, such as graupel formation, supercooled liquid water content and ice crystal collisions. ICEFLUX scheme, by explicitly modelling ice fluxes, provides a more realistic approach in predicting charge separation and lightning activity (Finney et al., 2014). This leads to improved flash rate estimations, particularly in midlatitude storms where vertical motion, ice microphysics and latent heat fluxes play a complex role in thunderstorm electrification.

**Changes in the manuscript:** Please see the modified lines 331–332, "While LNOx-ICEFLUX provides a reasonable estimate of flash rate magnitudes over both the tropics and mid-latitudes, it struggles to accurately capture the observed spatial pattern of lightning flashes, emphasizing the need for further improvement."

Also see the lines 310–313 added newly, "ICEFLUX scheme, by explicitly modelling ice fluxes, provides a more realistic approach in predicting charge separation and lightning activity (Finney et al., 2014). This leads to improved flash rate estimations, particularly in midlatitude storms where vertical motion, ice microphysics and latent heat fluxes play a complex role in thunderstorm electrification."

13. Table 2: Add rows showing corresponding values for ISS-LIS and LIS/OTD.

**Reply:** This is done.

**Changes in the manuscript:** Rows showing flash rate values from ISS-LIS and LIS/OTD are added in Table 2.

14. L304-308: The accuracy of the lightning data is also lower at high latitudes. As the data sets contain fewer years. Also, what fraction of flashes does ISS-LIS capture? Does it only view a given area for a few minutes each day? Data are also lower quality over the oceans due to sampling limitations.

**Reply:** Lightning Imaging Sensor (LIS) is mounted on the International Space Station (ISS) (domain: $\pm 55°$ latitudes). ISS operates in low Earth orbit (LEO) and overpasses one region on the earth surface up to three times a day and up to two times in the tropics. Lightning observation of a specific point lasts up to 90 seconds per overpass (Erdmann et al., 2020). The flash detection efficiency of ISS-LIS is around 60% with diurnal variability of 51%–75% (Blakeslee et al., 2020), as already mentioned in the manuscript (line number 236–237). Despite of its lower quality over the oceans, the lightning data provided by ISS-LIS is available for the year 2018, which is our study period, giving the opportunity to evaluate the simulated flash rates with respect to the ISS-LIS observations. We also have included comparison with respect to combined climatology product of satellite observations from the Optical Transient Detector (OTD) and the LIS, for the period of May, 1995 to December, 2014 (please see the lines 237–241).

**Changes in the manuscript:** We have included more details about ISS-LIS data. Please see the lines 234–236, "ISS operates in low Earth orbit (LEO) and overpasses one region on the earth surface up to three times a day and up to two times in the tropics. Lightning observation of a specific point lasts up to 90 seconds per overpass (Erdmann et al., 2020)."

15. L330: Are these correlations temporal or spatial? Based on the context and values, it appears they are temporal – correlations between monthly average flash rates for each latitude band?

**Reply:** The correlations are temporal. The statistical analysis presented in Table 4 are conducted for monthly mean flash rates from simulations in comparison to the monthly mean flash rates from satellite observations, for each latitude band. This is mentioned in lines 347–348, "The modelled monthly mean flash rates from the simulations exhibit a strong positive correlation with satellite observations (ISS-LIS and LIS/OTD), with correlation coefficients ranging from 0.85 to 0.97 (Table 4)." The word 'temporal' is now included in lines 351–352.

**Changes in the manuscript:** Please see the lines 351–352, "In contrast, a weaker negative temporal correlation is observed over tropical and mid-latitudinal oceans, indicating an inverse relationship between simulated and observed seasonality in flash rates in these regions."

16. L348: You hint at factors of 2-3 magnitudes but only show factors varying by factors of 2-4. Rephrase or give more details.

**Reply:** As per Tost et al. (2007), the scaling factors may vary up to 2–3 orders of magnitude to match observations, based on the lightning parameterization used and the results obtained. Tost et al. (2007) reported scaling factors as large as 435 and as low as 0.74 to match observations. A recent study by Finney et al. (2016) determined that the global flash rate scaling factors required for the UKCA model are 1.44 and 1.12 for the CTH and ICEFLUX lightning parameterizations, respectively. Another study by Gordillo-Vázquez et al. (2019) produce scaling factors 2.05 and 4, respectively for the CTH and ICEFLUX lightning schemes

in Community Atmosphere Model (CAM5). In our study, the flash rates from LNOx-ICEFLUX simulation, are adjusted by a factor 5 to reconcile with the satellite observations.

**Changes in the manuscript:** Please see the rephrased lines 367–369, "A study by Tost et al. (2007) reported that scaling factors may vary by up to 2–3 orders of magnitude, depending on the lightning parameterization used and the resulting flash rate, to better match the observations."

17. L350: Given the 4.9th power of cloud height variation, varying the minimum depth would have little effect.

**Reply:** A study by Luhar et al. (2021) has shown that increasing or decreasing the minimum cloud thickness value by 1 km from 5 km result in a change of $-3.2\%$ and $1.7\%$, respectively, in the modelled global flash rate, estimated using the CTH scheme. The effect may be lesser but still contributes to the uncertainty in estimating flash rates. Therefore, we mention that, it would be worthwhile to study the sensitivity of the modelled flash rates to the minimum cloud depth. We have moved these lines to the Section 2.2.1, where we introduced the minimum required cloud depth for the first time.

**Changes in the manuscript:** Please see the lines 125–128, "The assumption of minimum required cloud depth of 5 km, may introduce uncertainty in estimating lightning flashes, as it inherently assumes that every convective cloud with depth of 5 km corresponds to a thunderstorm (Luhar et al., 2021). It would be worthwhile to investigate the sensitivity of the modelled flash rates to the minimum cloud depth by varying this arbitrary threshold, either increasing or decreasing it."

18. L354: I'm not sure why you use "but" here. How did Finney determine "better"?

**Reply:** The study by Finney et al. (2014) showed that the correlations of the monthly average of upward ice flux at 340, 440 and 540 hPa, formed from ERA-Interim reanalysis for the grid cells over land, against LIS flash density were found stronger for lower pressure levels (340 and 440 hPa) in comparison to the higher pressure level (540 hPa), specifically over land. As per Finney et al. (2014), further investigation at that point was not necessary since only slight gains could be made by arbitrarily optimising the pressure level. We have now removed this line to avoid confusion.

**Changes in the manuscript:** We have removed the line.

19. L378: Remind the reader how you determined the vertical partitioning of LNOx emissions. Do these plots sum the contributions from IC and CG flashes? Do you assume only IC flashes for some altitudes and only CG for others etc.

**Reply:** We have added few lines reminding how we determined the vertical partitioning of LNOx emissions. The emissions from CG and IC flashes are calculated separately considering CG flashes only below the freezing level and the IC flashes only above the freezing level and below the cloud top. A simple vertical structure of the emissions is adopted in this study, considering the emissions to be evenly distributed over an altitude range.

**Changes in the manuscript:** Please see the lines 391–394, "The emissions from CG and IC flashes are calculated separately considering CG flashes only below the freezing level and the IC flashes only above the freezing level and below the cloud top. A simple vertical structure of the emissions is adopted in this study, considering the emissions to be evenly distributed over an altitude range."

20. L385: What previous studies are you referring to and are you certain they are plotting mass and not mixing ratio.

**Reply:** We have added the references of the previous studies (Pickering et al., 1998; Ott et al., 2010; Luhar et al., 2021). These studies plotted the LNOx emissions as percentage of total LNOx mass per km.

**Changes in the manuscript:** Please see the lines 401–403, "The vertical profiles available from previous studies, e.g., Pickering et al. (1998); Ott et al. (2010); Luhar et al. (2021), reveal a similar shape of all the profiles but contributing maximum at upper tropospheric region (within 2–4 km of the tropopause) rather than mid-troposphere."

**21. L393: "evenly distributed over an altitude range". This information should be given earlier in the text.**

**Reply:** The line has been shifted to provide the information earlier in the text.

**Changes in the manuscript:** Please see the lines 393–394, "A simple vertical structure of the emissions is adopted in this study, considering the emissions to be evenly distributed over an altitude range."

**22. L417-420: Midlatitude and polar region changes vary greatly by season. Thus, it is not surprising that annual mean changes are relatively small. What percent changes are seen in the late spring/summer.**

**Reply:** Yes, it is true that large seasonal changes are observed at mid-latitudes and polar region, lowering the annual mean increase in the $O_3$ mixing ratio due to inclusion of LNOx emissions. As mentioned in the manuscript, a moderate (3%–5%) to low (1%–2%) increase in annual mean of mid and upper tropospheric $O_3$ is observed over mid-latitudes followed by the polar region. The increase is comparatively higher during late spring and early summer (May–August) being 6%–15% over mid-latitudes and 2%–4% over polar regions.

**Changes in the manuscript:** Please see the newly added line 433–434, "The increase is comparatively higher during late spring and early summer (May–August) being 6%–15% over mid-latitudes and 2%–4% over polar regions."

**23. L438: Your model top may be too low to adequately resolve cross-tropopause transport.**

**Reply:** We acknowledge that the current model top (200 hPa) may limit the ability to fully capture cross-tropopause transport, especially in the upper troposphere and lower stratosphere. Future model improvements will focus on extending the model top to higher altitudes to better resolve these processes and improve the modelling of stratosphere-troposphere exchange.

**Changes in the manuscript:** Please see the modified line 454–456, "The underestimation suggests that the modeled stratosphere-troposphere exchange still requires significant refinement, and the cross-tropopause transport may not be adequately resolved due to the low model top."

**24. L485: What is causing that unobserved peak?**

**Reply:** The peak as observed at the latitude band 20°–30°N, of $1.75 \times 10^{15}$ molecules cm$^{-2}$, is due to the high $NO_2$ column density estimated from simulations over the southern and south-east Asia due to high $NO_2$ emissions from larger industrial activities, than that obtained from OMI observations. As we mentioned in the manuscript, this peak is not observed from OMI. On the other hand, a study by Luhar et al. (2021) has depicted that the $NO_2$ column density obtained from Copernicus Atmosphere Monitoring Service (CAMS) reanalysis data, shows a peak of $1.5 \times 10^{15}$ molecules cm$^{-2}$ at this latitude band (20°–30°N), where OMI underestimates the $NO_2$ column density. The higher uncertainty in OMI retrieved $NO_2$ columns, as compared with available satellite observations (GOME-2, SCIAMACHY and TROPOMI) is considerable in this regards. The uncertainties are primarily due to instrumental errors, limitations of the OMI sensor in capturing the $NO_2$ below the cloud level, vertical profile assumptions and surface reflectivity (Bucsela et al., 2013; Boersma et al., 2018).

**Changes in the manuscript:** Please see the modified lines 511–519, "The peak at 20°–30°N, of $1.75 \times 10^{15}$ molecules cm$^{-2}$, is due to the high $NO_2$ column density estimated from simulations over the southern and south-east Asia due to high $NO_2$ emissions from larger industrial activities. This peak is however not observed in OMI observations. On the other hand, a study by Luhar et al. (2021) has depicted that the $NO_2$ column density obtained from Copernicus Atmosphere Monitoring Service (CAMS) reanalysis data, shows a peak of $1.5 \times 10^{15}$ molecules cm$^{-2}$ at this latitude band (20°–30°N), where OMI underestimates the $NO_2$ column density. The higher uncertainty in OMI retrieved $NO_2$ columns, as compared with available satellite observations (GOME-2, SCIAMACHY and TROPOMI) is considerable in this regards. The uncertainties are primarily due to instrumental errors, limitations of the OMI sensor in capturing the $NO_2$ below the cloud level, vertical profile assumptions and surface reflectivity (Bucsela et al., 2013; Boersma et al., 2018)."

**25. L486: Higher uncertainty in OMI observations relative to what?**

**Reply:** The higher uncertainty in OMI retrieved $NO_2$ columns, as compared with available satellite observations (GOME-2, SCIAMACHY and TROPOMI), is primarily due to instrumental errors, limitations of the OMI sensor in capturing the $NO_2$ below the cloud level, vertical profile assumptions and surface reflectivity (Bucsela et al., 2013; Boersma et al., 2018).

**Changes in the manuscript:** Please see the modified lines 516–519, "The higher uncertainty in OMI retrieved $NO_2$ columns, as compared with available satellite observations (GOME-2, SCIAMACHY and TROPOMI) is considerable in this regards. The uncertainties are primarily due to instrumental errors, limitations of the OMI sensor in capturing the $NO_2$ below the cloud level, vertical profile assumptions and surface reflectivity (Bucsela et al., 2013; Boersma et al., 2018)."

**26. L553: How do you determine the $CH_4$ lifetime, i.e., what $CH_4$ distribution do you assume. Yes, a $CH_4$ lifetime of 4.5–4.8 years is quite short and yes your OH must have a considerable high-bias. Your mean concentrations of $\approx 15 \times 10^5$ are approximately 50% greater than the canonical value of $10 \times 10^5$ and as you note higher than the multi-model mean of $11 \times 10^5$.**

**Reply:** The $CH_4$ concentration is considered from chemical boundary conditions from CAMS reanalysis dataset of atmospheric compositions produced by ECMWF, as in our study, $CH_4$ anthropogenic emissions are not taken into account. This is mentioned in the lines 218–219, "[OH] is taken from the simulations in CHIMERE, whereas [$CH_4$] is from chemical boundary conditions derived from CAMS reanalysis dataset of atmospheric compositions, as $CH_4$ anthropogenic emissions are not taken into account in the model". The annual mean $CH_4$ burden (1930–1933 Tg) estimated in this study over the NH is $\approx 20\%$ lower than the multi-model mean $CH_4$ burden, obtained from ACCMIP simulations (Naik et al., 2013), considering half of the global $CH_4$ burden over the NH ($\approx 2406$ Tg). The annual mean OH concentration is also overestimated by 30%–38%, in comparison to the multi-model mean obtained from ACCMIP simulations (Naik et al., 2013) promoting the increased chemical loss of $CH_4$. Therefore, the lower $CH_4$ burden and higher chemical loss due to reaction with OH cause the underestimated lifetime of $CH_4$ in this study, even in the absence of LNOx. Therefore, the underestimated $CH_4$ lifetime in the present study is not attributed to LNOx but likely stems from other factors including issues related to deficiencies in $CH_4$ burden, the chemistry or photolysis schemes.

We have included these details in the manuscript.

[revised manuscript text omitted]

---

## Author Response (AR3)

**Review replies for the manuscript entitled "Representing improved tropospheric ozone distribution over the Northern Hemisphere by including lightning NOx emissions in CHIMERE" (egusphere-2024-3087) by Sanhita Ghosh et al.**

We sincerely thank the editor and the reviewer for their valuable time and insightful feedback on our manuscript. Below, we address each comment in detail and outline the corresponding revisions made to improve the manuscript.

**Reviewer 1**

1. L300-304: "which is not enough to differentiate land and ocean flashes" I don't understand the above sentence. If the ICEFLUX scheme underestimates oceanic flashes wouldn't that indicate that it is overestimating the land-ocean difference as opposed to being unable to differentiate the difference.

**Reply:** The reviewer is right. We have modified the sentance to avoid confusion.

According to Finney et al. (2014), the ICEFLUX scheme underestimates oceanic flash rates primarily due to weaker updraft strength in oceanic storms. This leads to less efficient charge separation, which in turn results in fewer lightning flashes. The ICEFLUX scheme explicitly relates lightning flash rates to the upward ice flux and since oceanic storms generally have weaker convection compared to land storms, the scheme naturally predicts lower flash rates over the ocean.

On the other hand, CTH scheme relies on cloud-top height as a proxy for lightning activity, which may not fully account for differences in updraft strength and charge separation efficiency over different surfaces, causing the estimation of relatively higher oceanic flash rates.

**Changes in the manuscript:** Please see the modified lines 298–300 in the manuscript, "The ICEFLUX scheme explicitly relates lightning flash rates to the upward ice flux; therefore, the weaker updraft strength in oceanic storms leads to less efficient charge separation, resulting in fewer lightning flashes over the ocean (Finney et al., 2014)."

2. L545-547: "The simulations also identify a secondary maximum in $NO_2$ column density at 35°–45°N, which aligns with satellite observations over the mid-latitudes, however, shows an underestimation from simulations." I don't understand what you mean here. Please rephrase.

**Reply:** We have modified the sentence for clarity.

**Changes in the manuscript:** Please see the modified lines 522–523 in the manuscript, "A secondary maximum in $NO_2$ column density is identified between 35°–45°N from simulations as well as from satellite observations. However, the simulated $NO_2$ column density is underestimated at mid-latitudes by 20%–40%."

**References**

Finney, D. L., Doherty, R. M., Wild, O., Huntrieser, H., Pumphrey, H. C., and Blyth, A. M.: Using cloud ice flux to parametrise large-scale lightning, Atmospheric Chemistry and Physics, 14, 12 665–12 682, https://doi.org/10.5194/acp-14-12665-2014, 2014.